# Genome-wide association study of placental weight identifies distinct and shared genetic influences between placental and fetal growth

**A list of authors and their affiliations appears at the end of the paper**

A well-functioning placenta is essential for fetal and maternal health throughout pregnancy. Using placental weight as a proxy for placental growth, we report genome-wide association analyses in the fetal ($n$ = 65,405), maternal ($n$ = 61,228) and paternal ($n$ = 52,392) genomes, yielding 40 independent association signals. Twenty-six signals are classified as fetal, four maternal and three fetal and maternal. A maternal parent-of-origin effect is seen near *KCNQ1*. Genetic correlation and colocalization analyses reveal overlap with birth weight genetics, but 12 loci are classified as predominantly or only affecting placental weight, with connections to placental development and morphology, and transport of antibodies and amino acids. Mendelian randomization analyses indicate that fetal genetically mediated higher placental weight is causally associated with preeclampsia risk and shorter gestational duration. Moreover, these analyses support the role of fetal insulin in regulating placental weight, providing a key link between fetal and placental growth.

The placental connection between fetus and mother provides nutrients and oxygen to the fetus while removing waste products from fetal blood. The placenta produces hormones, growth factors and cytokines, allowing maternal immunoglobulin G (IgG) antibodies to pass to the fetus, giving newborns innate immunity. Suboptimal placentation can lead to intrauterine growth restriction[1], miscarriage, preterm birth[2] and preeclampsia[3,4] A poorly functioning placenta is associated with risk of growth restriction[5], adverse neurodevelopment[6] and cardiometabolic diseases[7–11].

Placental weight (PW) is easily measured and is often used in epidemiological studies[12,13] to proxy placental growth and function. The placental-fetal growth nexus is reflected by a positive correlation ($r$ = 0.6) between placental and birth weight (BW)[12,14]. Genome-wide association studies (GWAS) have identified genetic loci in the maternal and fetal genomes associated with BW[15,16] being enriched for placental expression quantitative trait loci (eQTLs)[17]. However, no GWAS of PW is yet available, and the relationship between genetics of placental

growth, fetal growth and adverse pregnancy outcomes (for example, preeclampsia) remains unclear. Although placenta is primarily composed of cells with fetal origin, it is intricately connected to maternal physiology[18–20]. Genetic analyses offer the opportunity for insight into the complex interplay of direct fetal, indirect maternal and parent-of-origin effects (POEs), which we hypothesize underlie placental growth and function.

We conducted GWAS of PW in term, singleton pregnancies, meta-analyzing fetal, maternal or paternal genotype data from 21, 16 and six European studies, respectively (Fig. 1). Analyses of 19,861 child–mother–father trios with PW measurements enabled a better understanding of the relationship between fetal and maternal effects, including POE. We categorized loci according to their association with BW, examined genetic links between PW and pregnancy, perinatal and later-life outcomes and used Mendelian randomization (MR) to assess causal relationships between maternal and offspring characteristics and PW.

✉e-mail: stefan.johansson@uib.no; r.freathy@exeter.ac.uk; fee@ssi.dk; pal.njolstad@uib.no

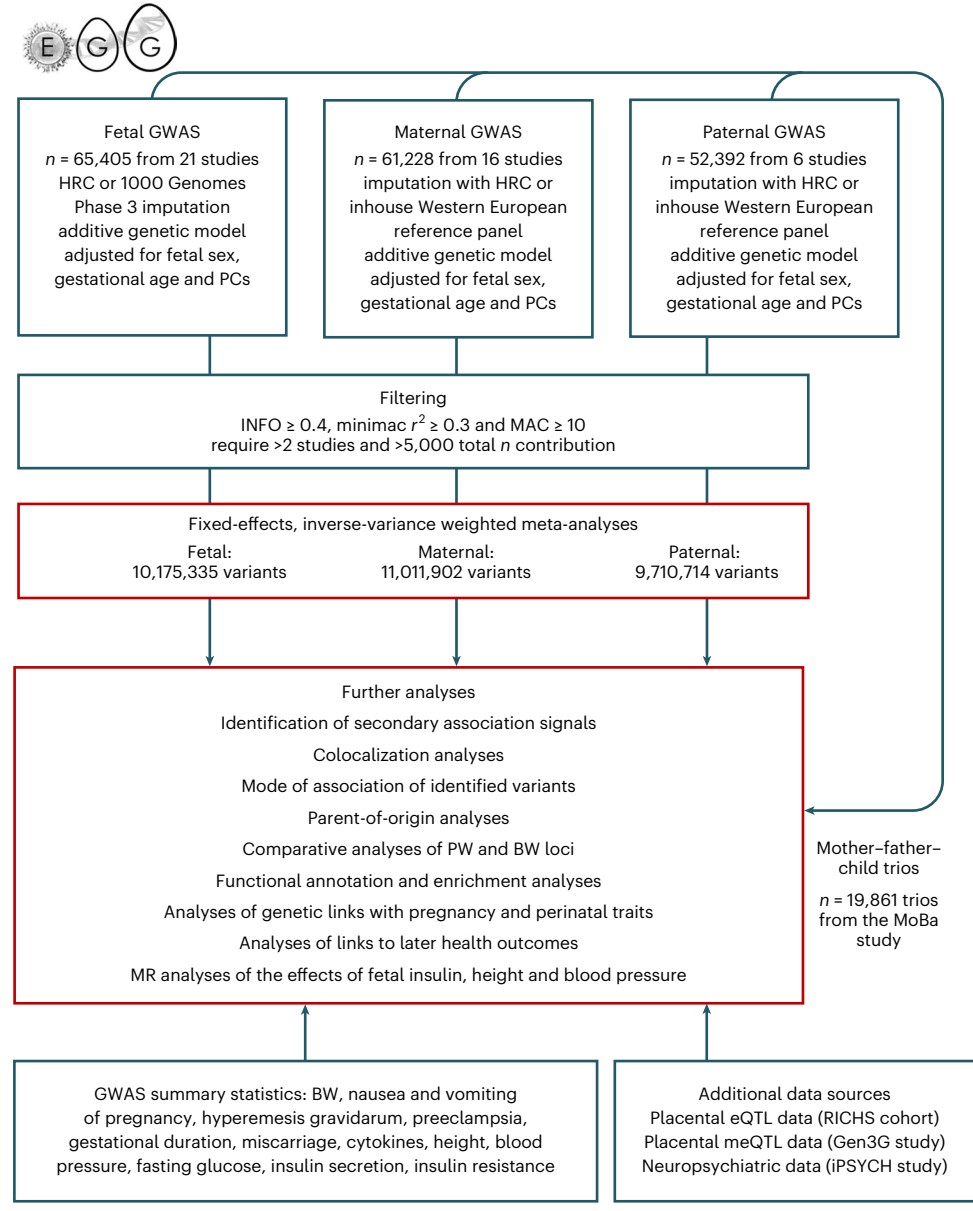

**Fig. 1 | Flow chart of the study design.** HRC, Haplotype Reference Consortium; MAC, minor allele count; PC, principal component.

## Results

### Meta-analyses of fetal, maternal and paternal GWAS

We performed GWAS meta-analyses of PW adjusted for fetal sex and gestational duration against fetal ($n = 65,405$), maternal ($n = 61,228$) and paternal genomes ($n = 52,392$; Fig. 1). Cohorts consisted of offspring, parents or both (Methods and Supplementary Tables 1–6 provide cohort information, data collection and genotyping). After data cleaning and imputation, 11 million SNPs were analyzed. The fetal GWA meta-analysis identified 32 independent loci at $P < 5 \times 10^{-8}$, the maternal analysis identified four and the paternal identified two loci (Fig. 2, Table 1, Supplementary Table 7 and Supplementary Fig. 1a–e (regional association plots by locus)). We found little evidence of heterogeneity among cohorts at any locus (Supplementary Table 7). Approximate conditional and joint analysis (COJO) further identified secondary association signals at three fetal loci (Methods; Table 1 and Supplementary Table 7). A comparison of effect sizes against minor allele frequencies for those 41 association signals was in line with expectations from statistical power (Extended Data Fig. 1). We also conducted analyses adjusted only for fetal sex (that is, not gestational age), which showed high correlations with our main results (all $r\_g \geq 0.99$). Four additional loci reached genome-wide significance in the fetal sex-adjusted analyses, all of which were close to genome-wide significance in our main fetal analysis (Supplementary Table 8).

### Fetal and parental contributions to association signals

While the fetal genotype may influence PW directly, the maternal genotype may have an effect via the intrauterine environment. To calculate the SNP heritability of the fetal, maternal and paternal contributions, we used a framework within genomic structural equation modeling (gSEM)[21,22]. We found substantial contributions from the fetal ($h^2 = 0.22$ (s.e. = 0.03)) and maternal ($h^2 = 0.12$ (s.e. = 0.02)) genomes to variation in PW, and a small component from the paternal genome ($h^2 = 0.06$ (s.e. = 0.02); Supplementary Table 9 and Extended Data Fig. 2), which may be due to nonadditive effects such as POE not accounted for in the model. We also found a genetic correlation between the three latent variables suggesting that the fetal effect on PW was negatively correlated with both maternal and paternal effects, conversely maternal and paternal effects were positively correlated (Supplementary Table 9).

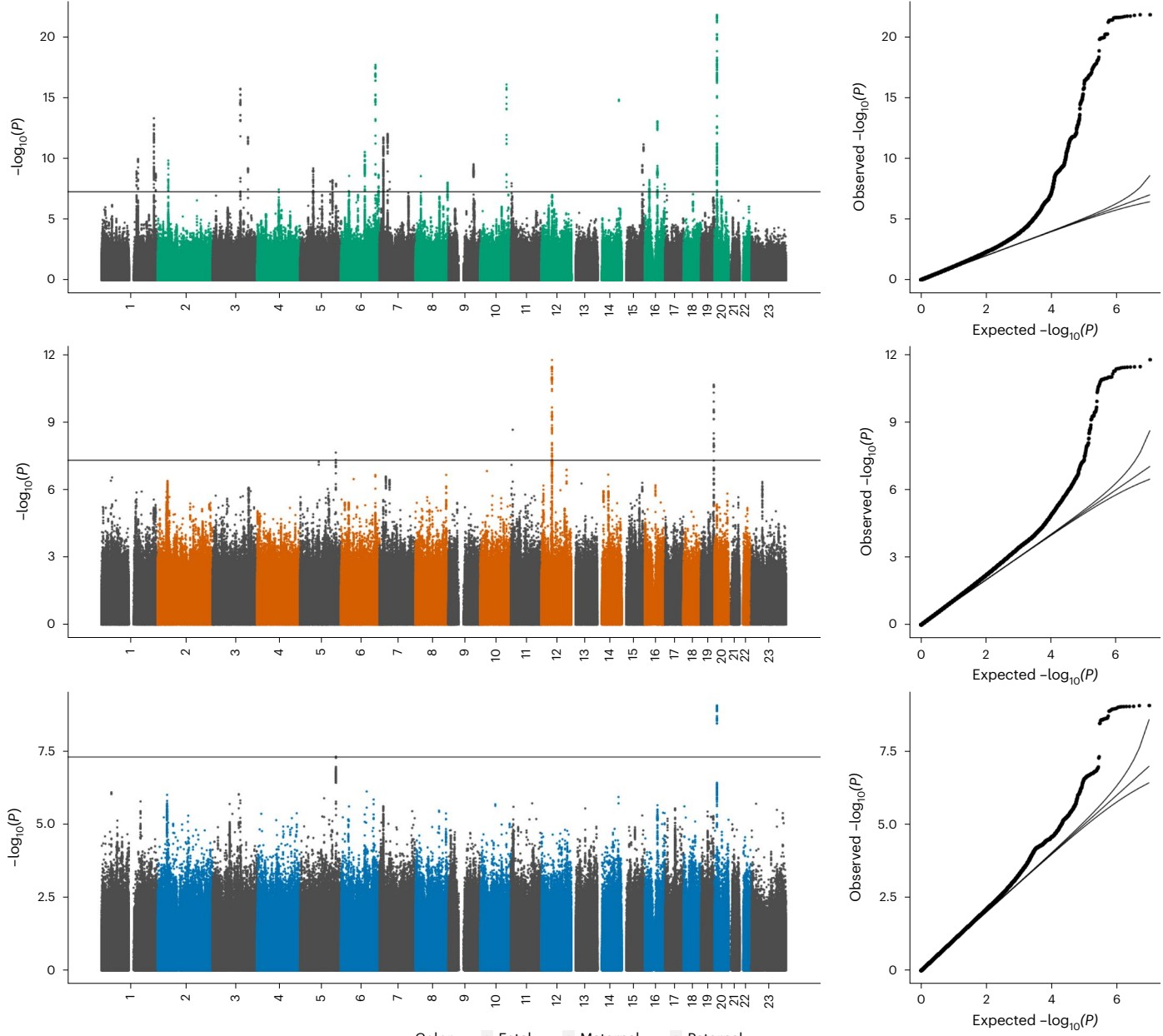

**Fig. 2 | Genome-wide association results for PW.** Manhattan plots of $-\log_{10}(P$ values) across the chromosomes and corresponding quantile–quantile plot of observed versus expected $-\log_{10}(P$ values) for meta-analyses of SNP associations with PW in the fetal GWAS (top, $n$ = 65,405 children), the maternal GWAS (middle, $n$ = 61,228 mothers) and the paternal GWAS (bottom, $n$ = 52,392 fathers).

To estimate fetal-specific, maternal-specific or paternal-specific effects on PW at all 41 identified loci, we applied a weighted linear model (WLM)[15] to the GWA meta-analysis summary statistics. The WLM can accurately approximate conditional effects in the absence of genotyped child–mother pairs (Supplementary Table 10). A total of 26 SNPs were classified as fetal-only, four maternal only, two fetal and maternal with effects in opposite directions and one fetal and maternal with effects in the same direction. We could not resolve the classification for the remaining eight loci (Supplementary Table 10 and Extended Data Figs. 3 and 4).

We next performed within-family analyses in the child–mother–father trio subset from the Norwegian Mother, Father and Child Cohort (MoBa)[23–25] (Supplementary Table 10). Conditional analyses showed good agreement with the mode-of-association categories based on WLM results. Using phased genotypes in MoBa children, we further decomposed the association signals into their mode of transmission

(that is, maternal transmitted, maternal nontransmitted, paternal transmitted and paternal nontransmitted alleles) and compared our results with a recent, similar analysis of BW[16] (Supplementary Table 11). For loci associated with both traits, effect size estimates and mode of transmission were consistent between BW and PW analyses (Fig. 3a and Supplementary Fig. 2). Among the eight unclassified loci after WLM analysis of PW, five were identified previously in BW GWAS and classified as fetal[16] and one was classified as fetal and maternal with the same direction of effect (Supplementary Table 11).

For the signal identified in the paternal GWAS near *EBF1* (rs75512885), we could not resolve the mode of association using WLM or offspring–parent trio analyses, but this variant is in moderate linkage disequilibrium (LD) with rs72813918 ($r^2$ = 0.57), classified previously as fetal for BW[16] (Supplementary Table 11). The lead SNP of the other paternal GWAS locus (rs2207099, near *LOC339593*) colocalized with the nearby fetal lead SNP (rs6040436; posterior probability for shared

**Table 1 | Variants associated with PW (adjusted for sex and gestational duration) in the fetal, maternal and paternal GWAS meta-analyses**

| Locus name | Chr | Position (b37) | rsID | EA | OA | Effect allele frequency | $\beta$ | s.e. | P | n | Maternal–fetal classification | PW/BW classification |
|---|---|---|---|---|---|---|---|---|---|---|---|---|
| | | | | | | Fetal | | | | | | |
| DCST2 | 1 | 155001281 | rs150138294 | A | G | 0.053 | 0.080 | 0.013 | $1.07\times10^{-9}$ | 65,405 | Unclassified | PW and BW same direction |
| RPL31P11 | 1 | 161651064 | rs723177 | T | C | 0.296 | 0.039 | 0.006 | $1.02\times10^{-10}$ | 65,405 | Fetal | Predominantly or only PW |
| TSNAX-DISC1, LINC00582 | 1 | 231733795 | rs1655296 | G | T | 0.393 | 0.042 | 0.006 | $4.86\times10^{-14}$ | 65,405 | Fetal | Predominantly or only PW |
| TSNAX-DISC1, LINC00582* | 1 | 231794081 | rs140691414 | T | C | 0.007 | 0.261 | 0.039 | $2.54\times10^{-11}$ | 57,158 | Fetal | PW and BW same direction |
| CHRM3 | 1 | 239822859 | rs10925945 | T | C | 0.936 | 0.067 | 0.011 | $1.69\times10^{-9}$ | 65,405 | Unclassified | Predominantly or only PW |
| EPAS1 | 2 | 46567276 | rs4953353 | G | T | 0.622 | 0.042 | 0.007 | $1.38\times10^{-10}$ | 48,809 | Unclassified | PW and BW same direction |
| ADCY5 | 3 | 123065778 | rs11708067 | G | A | 0.233 | 0.053 | 0.007 | $1.83\times10^{-16}$ | 65,405 | Fetal and maternal | PW and BW same direction |
| LOC339894/ CCNL1 | 3 | 156795414 | rs9817452 | G | T | 0.614 | 0.040 | 0.006 | $1.72\times10^{-12}$ | 65,405 | Fetal | PW and BW same direction |
| PDLIM5 | 4 | 95531563 | rs74457440 | A | G | 0.233 | 0.036 | 0.007 | $3.37\times10^{-8}$ | 65,405 | Unclassified | PW and BW same direction |
| ACTBL2 | 5 | 57073666 | rs7722058 | T | C | 0.845 | 0.047 | 0.008 | $6.06\times10^{-10}$ | 65,405 | Fetal | PW and BW same direction |
| HSPA4 | 5 | 132444128 | rs72801474 | A | G | 0.105 | 0.055 | 0.009 | $7.34\times10^{-9}$ | 65,405 | Fetal | Predominantly or only PW |
| ARHGAP26 | 5 | 142429811 | rs3822394 | C | A | 0.263 | 0.036 | 0.006 | $5.80\times10^{-9}$ | 65,405 | Fetal | Predominantly or only PW |
| EBF1 | 5 | 158433339 | rs67265526 | T | C | 0.628 | 0.032 | 0.006 | $1.12\times10^{-8}$ | 65,405 | Fetal | Predominantly or only PW |
| NUDT3 | 6 | 34237188 | rs541641049 | A | G | 0.018 | 0.193 | 0.035 | $4.52\times10^{-8}$ | 55,263 | Fetal | PW and BW same direction |
| FKBP5/ MAPK13/TEAD3 | 6 | 35517390 | rs9800506 | T | G | 0.410 | 0.033 | 0.006 | $2.58\times10^{-9}$ | 65,405 | Unclassified | PW and BW same direction |
| HACE1 | 6 | 105130521 | rs12529634 | C | T | 0.872 | 0.056 | 0.008 | $2.85\times10^{-11}$ | 65,405 | Fetal | PW and BW same direction |
| ESR1 | 6 | 152042413 | rs11756568 | A | T | 0.703 | 0.052 | 0.006 | $1.95\times10^{-18}$ | 65,405 | Fetal | PW and BW same direction |
| PDE10A | 6 | 166182483 | rs1021508 | C | T | 0.633 | 0.034 | 0.006 | $2.09\times10^{-9}$ | 65,405 | Fetal | PW and BW same direction |
| PDE10A* | 6 | 166199513 | rs6456014 | C | A | 0.582 | 0.030 | 0.006 | $4.42\times10^{-8}$ | 65,405 | Fetal | Predominantly or only PW |
| ISPD | 7 | 16193877 | rs7783810 | C | T | 0.422 | 0.039 | 0.006 | $1.85\times10^{-12}$ | 65,405 | Fetal | PW and BW same direction |
| TBX20 | 7 | 35282931 | rs10486660 | A | C | 0.608 | 0.040 | 0.006 | $9.49\times10^{-13}$ | 65,405 | Fetal | PW and BW same direction |
| YKT6 | 7 | 44246271 | rs138715366 | C | T | 0.991 | 0.202 | 0.035 | $7.91\times10^{-9}$ | 63,786 | Fetal and maternal | PW and BW same direction |
| ENTPD4 | 8 | 23342043 | rs6557677 | T | A | 0.166 | 0.044 | 0.007 | $2.66\times10^{-9}$ | 65,405 | Fetal | Predominantly or only PW |
| SLC45A4 | 8 | 142247979 | rs12543725 | G | A | 0.585 | 0.032 | 0.006 | $9.20\times10^{-9}$ | 65,405 | Unclassified | PW and BW same direction |
| KLF4 | 9 | 110822658 | rs1434836 | A | G | 0.575 | 0.035 | 0.006 | $2.87\times10^{-10}$ | 65,405 | Fetal | PW and BW same direction |
| ADRB1 | 10 | 115805056 | rs1801253 | C | G | 0.743 | 0.052 | 0.006 | $8.06\times10^{-17}$ | 65,405 | Fetal | PW and BW same direction |
| KCNQ1 | 11 | 2839751 | rs2237892 | T | C | 0.056 | 0.077 | 0.014 | $1.03\times10^{-8}$ | 48,809 | Fetal and maternal | Predominantly or only PW |
| SERPINA1 | 14 | 94838142 | rs112635299 | T | G | 0.024 | 0.149 | 0.019 | $1.42\times10^{-15}$ | 64,541 | Fetal | Predominantly or only PW |

**Table 1 (continued) | Variants associated with PW (adjusted for sex and gestational duration) in the fetal, maternal and paternal GWAS meta-analyses**

| Locus name | Chr | Position (b37) | rsID | EA | OA | Effect allele frequency | β | s.e. | P | n | Maternal–fetal classification | PW/BW classification |
|---|---|---|---|---|---|---|---|---|---|---|---|---|
| *FES/FURIN* | 15 | 91428636 | rs7177338 | A | G | 0.536 | 0.035 | 0.006 | 3.28×10⁻¹⁰ | 65,405 | Unclassified | PW and BW same direction |
| *NR2F2* | 15 | 96852638 | rs55958435 | A | G | 0.713 | 0.042 | 0.006 | 6.82×10⁻¹² | 65,405 | Fetal | PW and BW same direction |
| *GPR139/ GPRC5B* | 16 | 20006986 | rs57790054 | G | A | 0.302 | 0.035 | 0.006 | 5.54×10⁻⁹ | 65,405 | Fetal | PW and BW same direction |
| *SLC6A2* | 16 | 55717569 | rs11866404 | C | G | 0.533 | 0.042 | 0.006 | 8.55×10⁻¹⁴ | 65,405 | Fetal | Predominantly or only PW |
| *SLC7A5* | 16 | 87882209 | rs876987 | G | A | 0.306 | 0.038 | 0.007 | 1.24×10⁻⁸ | 48,809 | Fetal | PW and BW opposite directions |
| *LOC339593* | 20 | 11200008 | rs6040436 | T | C | 0.444 | 0.054 | 0.006 | 1.55×10⁻²² | 65,405 | Fetal | PW and BW same direction |
| *LOC339593** | 20 | 11428113 | rs6078190 | C | A | 0.319 | 0.040 | 0.006 | 1.36×10⁻¹¹ | 65,405 | Fetal | Predominantly or only PW |
| Maternal | | | | | | | | | | | | |
| *EBF1* | 5 | 157808173 | rs72804545 | A | T | 0.888 | 0.064 | 0.011 | 2.23×10⁻⁸ | 41,211 | Maternal | PW and BW same direction |
| *LMO1* | 11 | 8255408 | rs2168101 | C | A | 0.696 | 0.047 | 0.008 | 2.13×10⁻⁹ | 41,356 | Maternal | PW and BW same direction |
| *SLC38A4* | 12 | 47180370 | rs180435 | C | G | 0.190 | 0.051 | 0.007 | 1.68×10⁻¹² | 61,153 | Maternal | PW and BW same direction |
| *NLRP13* | 19 | 56423893 | rs303998 | G | A | 0.404 | 0.039 | 0.006 | 2.12×10⁻¹¹ | 61,063 | Maternal | PW and BW same direction |
| Paternal | | | | | | | | | | | | |
| *EBF1* | 5 | 158317602 | rs75512885 | T | A | 0.925 | 0.080 | 0.015 | 4.96×10⁻⁸ | 31,714 | Unclassified | PW and BW same direction |
| *LOC339593* | 20 | 11207949 | rs2207099 | A | G | 0.448 | 0.038 | 0.006 | 8.77×10⁻¹⁰ | 52,271 | Fetal | PW and BW same direction |

The lead SNP is given in the rsID column. Where PW loci fell within 500 kb of BW loci, locus names were given the name of the BW locus in ref. 15; otherwise, the locus was named by the nearest gene. Secondary signals at a locus are marked with an asterisk. Effect alleles are coded to correspond to increasing PW. β and s.e. are given in s.d. units of PW. The subheadings under locus name indicate the GWAS analysis that the signal was identified in, and the two last columns classify signals into fetal and/or maternal mode of association (using a weighted linear model[15]) and into whether effects were only/predominantly on PW, or also on BW. Statistical tests are from linear regression, association results are two-sided and SNPs with $P<5×10^{-8}$ were considered associated. Chr, chromosome; EA, effect allele; OA, other allele; rsID, reference SNP cluster ID.

association = 0.97; Supplementary Table 7), hence was excluded from subsequent analyses, leaving 40 independent association signals.

Only one of the 40 lead SNPs was located near imprinted genes (Supplementary Table 11). The association with PW at rs2237892, in intron 10 of *KCNQ1*, was conferred by the maternally transmitted allele (Fig. 3b and Supplementary Fig. 2), consistent with the known maternal-only expression of *KCNQ1* and nearby *CDKN1C* genes. This variant shows POE on type 2 diabetes risk[26], where the maternally inherited risk allele (C) corresponds to the maternally inherited PW-decreasing allele identified here. Another independent nearby variant, rs234864, has been reported[16] as a maternally transmitted effect only for BW (Fig. 3c). The PW variant rs2237892 showed directionally consistent, but weaker evidence of association with BW in MoBa (Fig. 3d). This is consistent with the lower fetal effect size of rs2237892 on BW in previous BW GWAS[15] compared with the effect on PW (Supplementary Table 11).

**Correlations between PW and BW**

We confirmed the strong phenotypic correlation reported between PW and BW[12,14] in MoBa (Spearman's $r = 0.59$, adjusted for sex and gestational duration). We applied LD score regression[27] to estimate genetic correlations with published BW-association summary statistics[15], analyzing both main GWAS summary statistics, and WLM-adjusted

estimates of the conditional fetal, maternal and paternal effects (Fig. 4). Using WLM-adjusted effects for both PW and BW, the fetal and maternal genetic correlations between PW and BW remained strong. Interestingly, WLM-adjusted fetal effects on BW were negatively correlated with WLM-adjusted maternal effects on PW (Fig. 4c), suggesting that fetal genetic influences that raise BW correlate with opposing effects of the maternal genome that reduce PW. However, this observation could also be the result of collider bias, or the known negative correlation between maternal and fetal effect sizes induced by conditional analysis[15].

Of 40, 28 independent PW signals were also reported for BW or had a BW-lead SNP nearby (Supplementary Table 7). Colocalization analysis suggested that 19 of these represent a shared underlying association (posterior probability for shared signal >0.8), five signals were distinct (posterior probability for separate signals >0.8) and the final four loci were uncertain (both posterior probabilities <0.8).

Given the large proportion of signals colocalizing with BW loci, we aimed to distinguish loci showing associations with both PW and BW from those showing only (or predominantly) association with PW, by comparing PW and BW effect estimates (Supplementary Table 7; Methods). Twelve signals were classified as only or predominantly PW and 28 as both PW and BW signals, with one (near *SLC7A5*) showing opposite directions of effect, both with $P < 0.05$. The results obtained from WLM-adjusted analysis were consistent (Fig. 5, Extended Data

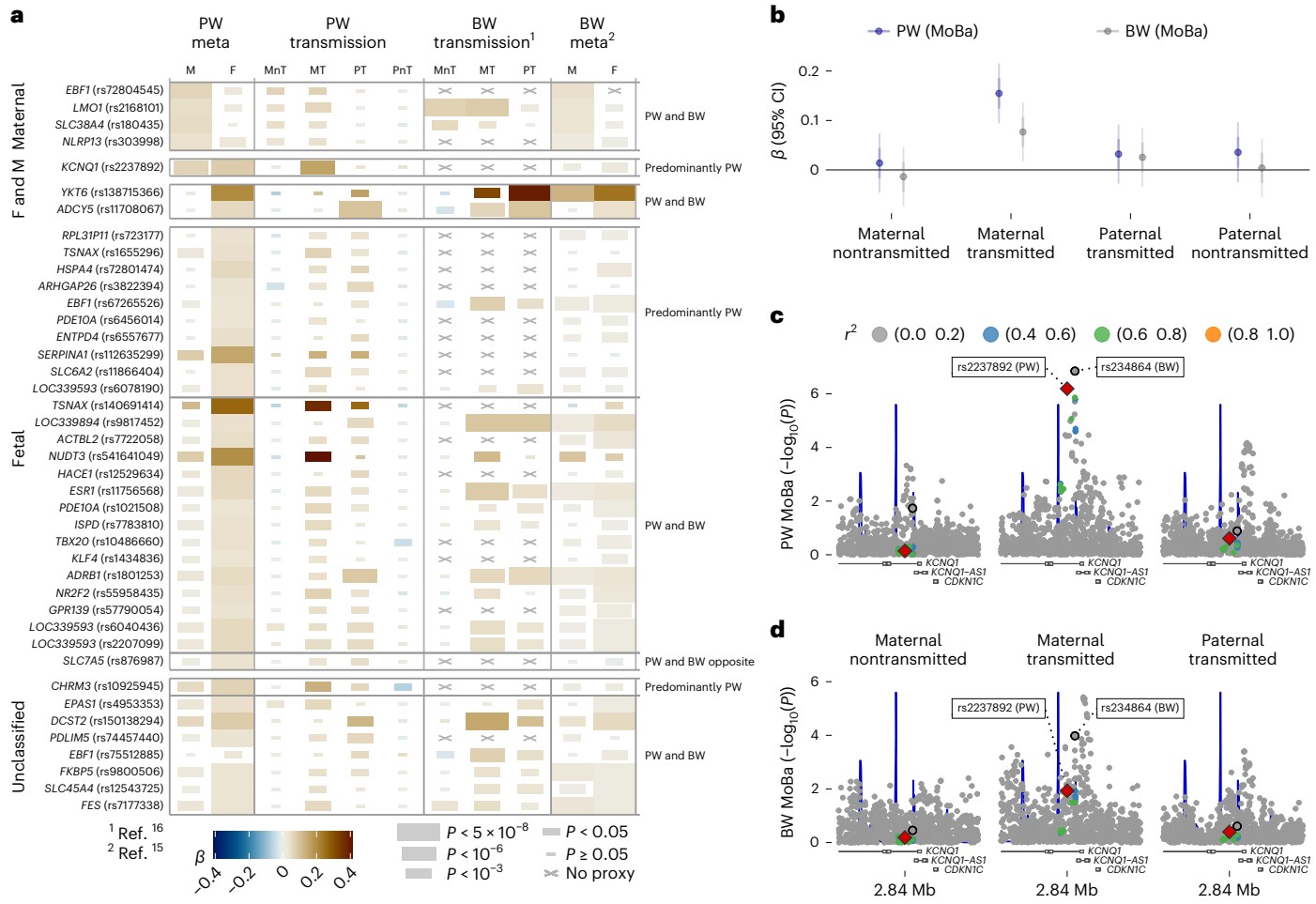

**Fig. 3 | Resolving fetal and parental contributions to PW associations.**
**a**, Effect size and significance estimates for the 41 association signals in our PW
meta-analysis (M: maternal genome, $n = 61,228$; F: fetal genome, $n = 65,405$;
not conditioned on each other), our PW allele contribution analysis in MoBa
($n = 19,861$), the BW allele contribution analysis discussed in ref. 16 (a cross
indicates that no genome-wide significant hit discussed in ref. 16 had $r^2 \geq 0.2$
with the PW-lead SNP) and the BW meta-analysis discussed in ref. 15 (M: maternal
genome, F: fetal genome, not conditioned against each other, a cross indicates
that no proxy was found with $r^2 \geq 0.2$). **b**, Effect size estimates for standardized
PW and BW adjusted for sex and gestational duration for rs2237892 near *KCNQ1*
in MoBa, $n = 19,861$. Thin and thick error bars, respectively, represent 95%
confidence intervals and one s.e. on each side of the point representing the effect

size estimate. **c,d**, Regional plots at the *KCNQ1* locus (chromosome 11, 2689751–
3007297 Mb), plotting $-\log_{10}$ of the unadjusted $P$ value against genomic location,
for the maternal nontransmitted, maternal transmitted, paternal transmitted
and paternal nontransmitted alleles in their association with standardized PW
(**c**) and BW (**d**) adjusted for sex and gestational duration in MoBa, $n = 19,861$.
Recombination rates are plotted in blue, and the exons of *KCNQ1*, *KCNQ1-AS1*
and *CDKN1C* are displayed under the plot. The leading variants for PW and BW,
rs2237892 and rs234864, are annotated with a diamond and a circle, respectively.
Points are colored according to their LD with rs2237892. F and M, fetal and
maternal; MnT, maternal nontransmitted allele; MT, maternal transmitted allele;
PT, paternal transmitted allele; PnT, paternal nontransmitted allele.

Fig. 5 and Supplementary Table 7). Thus, 12 of the 40 independent
signals only influence PW or have effects on PW that are larger than
any estimated effect on BW (Table 1).

**Connections to placental development and function**
We selected the 31 protein-coding genes closest to the index SNP of
the 32 loci identified in the fetal GWAS (that is, excluding *LOC339593*;
Table 1) and examined their expression using tissue-specific mRNA
abundance data from the Human Protein Atlas and a scRNA-seq dataset
of over 70,000 single cells at the decidual–placental interface in early
pregnancy (Methods). Among 61 tissues, the 31 genes ranked higher in
terms of mRNA abundance than other genes in placenta ($P = 1.8 \times 10^{-4}$;
Extended Data Fig. 6 and Supplementary Table 12). In the single-cell
analysis among the 32 different cell types at the early maternal–fetal
interface, expression of the 31 genes ranked higher than other genes in
cell types of fetal origin, including fetal endothelial cells ($P = 6.2 \times 10^{-5}$)
and syncytiotrophoblasts ($P = 1.9 \times 10^{-4}$), and also ranked higher in

maternal innate lymphocyte cells ($P = 1.2 \times 10^{-3}$; Extended Data Fig. 7
and Supplementary Table 12).

We then queried lead PW SNPs (and their proxies $r^2 \geq 0.8$) against
eQTL data from placenta in the RICHS dataset[28]. Lead PW SNPs tagged
placental eQTLs at four loci (*HSPA4, TBX20, SLC7A5* and *JAG1*; Sup-
plementary Table 7 and Extended Data Fig. 7). We additionally inves-
tigated whether any of the 40 independent lead SNPs were associated
with placental DNA methylation at CpG sites within 0.5 Mb. Using
Gen3G data[29,30], we identified placental methylation quantitative trait
loci (meQTL) at 21 of the 40 independent signals (false discovery rate
(FDR) ≤ 0.05). Among the 21 lead SNPs with identified meQTLs, indi-
vidual SNPs were associated with placental DNA methylation at up
to 15 CpG sites (Supplementary Fig. 3). Supplementary Table 7 shows
meQTLs with the lowest $P$ value for each lead SNP.

We performed further lookups of lead PW SNPs in available GWAS
of gene expression, plasma protein levels, diseases and traits (Sup-
plementary Tables 7, 13–15). These implicated several candidate genes

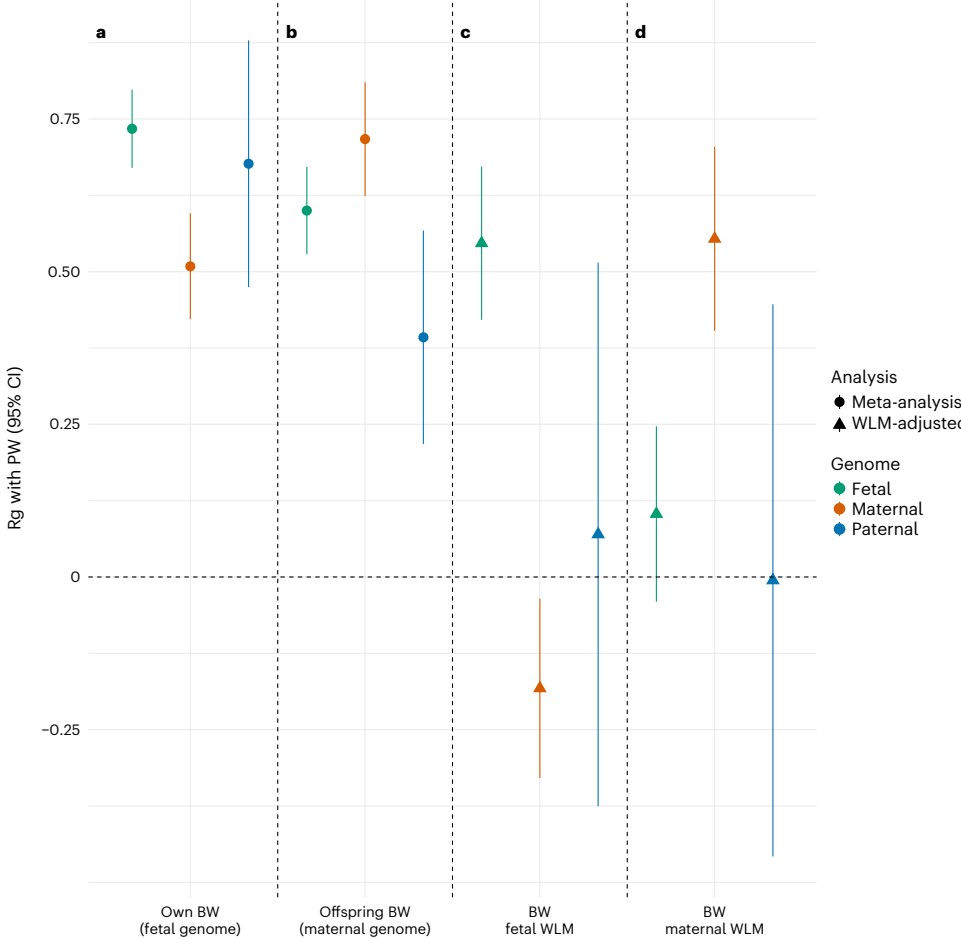

**Fig. 4 | Genetic correlation between PW and BW. a–d,** Genetic correlation estimates and corresponding 95% CIs between PW (current study) and BW ($n$ = 321,223 fetal and 230,069 maternal) from ref. 15. **a,** Fetal GWAS of BW.

**b,** Maternal GWAS of offspring BW. **c,** WLM-adjusted fetal GWAS of BW. **d,** WLM-adjusted maternal GWAS of BW. Values are provided for fetal, maternal and paternal effects before and after WLM adjustment. Rg, genetic correlation.

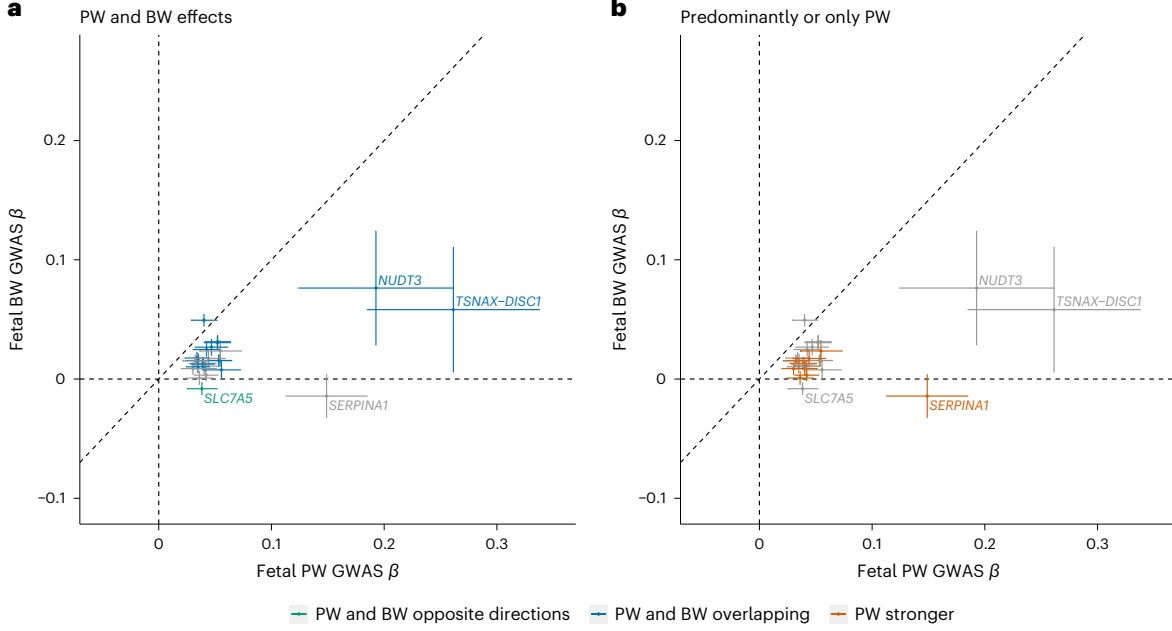

**Fig. 5 | Scatter plots comparing effect sizes from PW and BW GWAS for PW SNPs. a,b,** Scatter plots comparing effect size estimates and 95% confidence estimates from the PW GWAS ($n$ = 65,405) with those from the BW GWAS

($n$ = 321,223)[15]. **a,** Colored (blue and green) points represent variants classified as having both PW and BW effects. **b,** Colored (orange) points represent variants classified as having predominantly or only PW effects.

and potential biological insights. For example, the PW-raising allele of rs723177 at fetal *RPL31P11* (Supplementary Fig. 1a) is an eQTL and pQTL for *FCGR2A* and *FCGR2B* (receptors for the Fc region of IgG complexes) and associates with higher plasma protein levels (Supplementary Tables 14 and 15), suggesting a role in maternal antibody transfer across placenta[31]. The lead variant rs723177 is also the strongest placental meQTL in the region at the CpG site cg27514565 (overlapping DNase I hypersensitive sites (DHS) for primitive/embryonic and myeloid/erythroid tissues) located between *FCGR2C* and *FCGR3B*, and two meQTL (at cg14354529 and cg15531363) overlapping with trophoblast-specific DHS located downstream of *FCRLA* and *FCRLB* (Supplementary Fig. 3a). Associations at lead SNP rs10486660, at another fetal locus, suggest a role for this in placental morphology—the SNP is intragenic in the T-box transcription factor gene, *TBX20* (Supplementary Fig. 1b), and marks an eQTL for *TBX20* in placenta (Supplementary Table 7). Approximately 50% of *TBX20* placental expression is found in trophoblast cells[32], and *TBX20*-null mice show abnormal placental morphology compared to wild-type mice[33]. The lead SNP rs112635299 is in LD ($r^2 = 1.0$) with rs28929474, a missense variant in the serpin peptidase inhibitor gene *SERPINA1* (Supplementary Fig. 1c). The SNP associates with several traits, including levels of >20 circulating plasma proteins[34] (Supplementary Table 15) and causes autosomal recessive α-1 antitrypsin deficiency[35]. Within placenta, *SERPINA1* expression regulates expression of inflammatory cytokines and serine protease *HTRA1*-induced trophoblast invasion through induction of endoplasmic reticulum stress[34]. Finally, lead fetal SNP rs876987, the only variant showing an inverse effect on PW relative to BW, is in an intron of *SLC7A5*, which encodes a sodium-independent, high-affinity amino acid exchanger over membranes of several organs, responsible for uptake of essential amino acids in placenta[36]. This variant is an eQTL for *SLC7A5* in placenta and also a meQTL (Supplementary Table 7 and Supplementary Fig. 3d).

## PW and pregnancy complications

We tested whether the 40 independent lead SNPs were associated with pregnancy and perinatal traits, using GWAS summary statistics for nausea and vomiting of pregnancy or hyperemesis gravidarum[37], preeclampsia[38], gestational duration[39], miscarriage (recurrent and spontaneous)[40] and ten cytokines assayed from neonatal blood spots[41]. The SNPs showed more associations with nausea and vomiting of pregnancy (maternal genotype effects), and with gestational duration, preeclampsia and neonatal immunoglobulin A (IgA) levels (fetal genotype effects) than expected under the null distribution (Supplementary Fig. 4 and Supplementary Table 16)[39,40]. Scatter plots of effect sizes (Supplementary Fig. 5) suggested that fetal alleles predisposing to higher PW tended to associate with higher odds of preeclampsia and a shorter gestational duration. To test the effect of fetal PW-raising alleles on preeclampsia and gestational duration, we performed MR analyses. These showed that fetal genetic predisposition to a higher PW raises the risk of preeclampsia (odds ratio (OR) = 1.72 (95% confidence interval (CI): 1.19–2.47) per 1 s.d. higher fetal genetically predicted PW, $P = 6 \times 10^{-3}$) and shortens gestational duration (1.9 d (95% CI: 0.74–3.12) shorter per 1 s.d. higher fetal genetically predicted PW, $P = 3 \times 10^{-3}$; Supplementary Table 17 and Extended Data Fig. 8). We were unable to test causality between fetal genetically predicted PW and nausea and vomiting of pregnancy or IgA because only maternal effect estimates were available for the nausea and vomiting of pregnancy outcome[37] and only unadjusted fetal effect estimates for IgA[41]. The gestational duration results were similar to those seen for BW[42], where fetal genetic predisposition to higher BW is associated with shorter gestation, implying a general effect of fetoplacental growth[43]. However, known BW-associated SNPs[15] with fetal-only effects showed little evidence of association in the fetal GWAS of preeclampsia (Supplementary Fig. 4t), contrary to PW-associated SNPs with fetal-only effects (Supplementary Fig. 4u). MR analysis did not support a causal relationship between fetal genetic predisposition to higher BW and odds of preeclampsia ($P = 0.6$; Supplementary Table

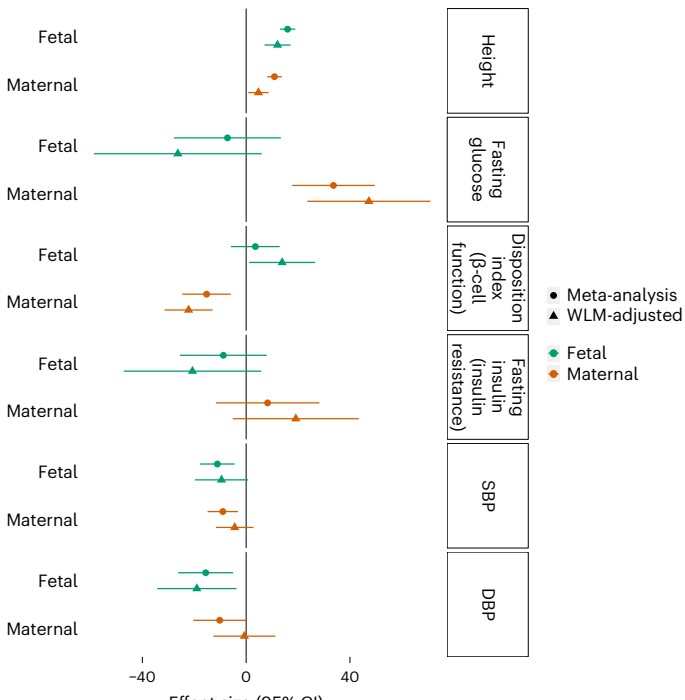

**Fig. 6 | MR analyses testing effects of maternal exposures (and fetal genetic predisposition to those exposures) on PW.** Point (effect size estimate) and line (95% CI of effect size) of two-sample MR analyses estimating the causal effects of maternal height, glycemic traits and blood pressure on PW, as well as the effects of fetal genetic predisposition to the same traits on PW. Maternal WLMs are adjusted for the effects of the fetal genotypes, and fetal WLM analyses are adjusted for the effects of the maternal genotypes. WLM, weighted linear model; SBP, systolic blood pressure; DBP, diastolic blood pressure. Units for exposures are height (s.d.), fasting glucose (s.d.), disposition index (s.d.), fasting insulin (s.d.), SBP (10 mmHg) and DBP (10 mmHg). Causal estimates are grams of PW per unit exposure.

17 and Extended Data Fig. 8), suggesting that increase in preeclampsia risk is more specific to placental growth.

## Maternal and fetal traits and effects on PW

We selected the following key maternal and fetal traits that influence BW[15,16] and used genetic instruments to assess whether there were similar causal relationships with PW: (1) maternal glucose and fetal insulin, (2) genetic predisposition to adult height and (3) blood pressure.

Fetal insulin is a major determinant of fetal growth. Lower fetal insulin secretion due to rare fetal *GCK* mutations leads to reduced BW and PW compared with siblings without mutations[44,45]. To investigate the role of fetal insulin on PW in the general population, we used the following three SNP sets in MR analyses: (1) 33 fasting glucose SNPs (maternal glucose crosses the placenta stimulating fetal insulin secretion); (2) 18 insulin disposition index SNPs (estimated by insulin secretion multiplied by insulin sensitivity, a β-cell function proxy) and (3) 53 BMI-adjusted fasting insulin SNPs (insulin resistance proxy in adults; Supplementary Table 18)[15]. We used WLM estimates to ensure maternal effects were adjusted for fetal effects, and vice versa. A genetic instrument representing 1 s.d. (0.4 mmol l⁻¹) higher maternal fasting glucose level was associated with a 47.4 g (95% CI: 23.7–71.2 g) higher PW ($P = 4.46 \times 10^{-4}$; Fig. 6 and Supplementary Table 18). In addition, a 1-s.d. genetically higher fetal disposition index was associated with a 13.9-g (1.2–26.6) higher PW. Alleles that raise insulin secretion tend to lower glucose levels, and this was reflected in the opposite direction of the effect estimates for maternal disposition index (Fig. 6 and Supplementary Table 18). The causal effect estimate for fetal insulin

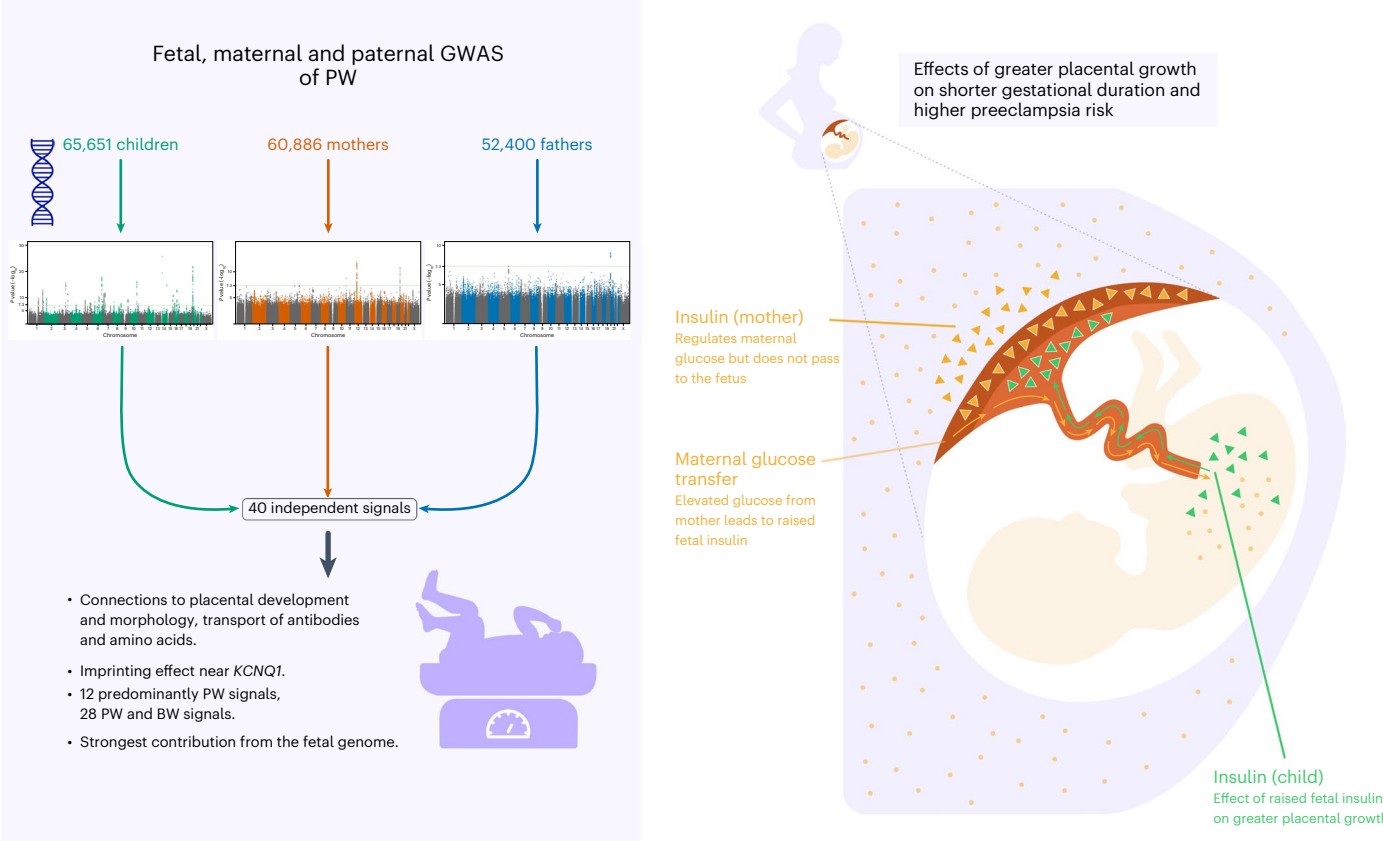

**Fig. 7 | Overview of main study findings.** Summary of results and conclusions from fetal, maternal and paternal meta-analyses (left panel) and a schematic of the results of Mendelian randomization analyses (right panel).

resistance was directionally consistent with causing lower PW, but the 95% CI crossed the null. These analyses support a role for insulin produced by the fetus, either directly or indirectly, in regulating the growth of the placenta, providing a key link between fetal insulin and placental growth.

Our analyses showed that fetal and maternal genetic predisposition to a greater adult height was associated with greater PW, consistent with previous associations with higher BW and length[15,16] (Fig. 6 and Supplementary Table 18). These findings further emphasize the contributions of common fetal genetic factors to the close relationship between fetal and placental growth.

Higher maternal blood pressure in pregnancy is associated with reduced fetal growth[46]. Using two-sample MR analyses, we found no evidence of a causal effect of either maternal systolic blood pressure (SBP; $P = 0.23$) or maternal diastolic blood pressure (DBP; $P = 0.91$; Fig. 6 and Supplementary Table 18). However, analyses using direct fetal genotype effects (adjusted for maternal genotype effects) suggested that a 10-mmHg genetically higher DBP in the fetus caused a 19.0-g (95% CI: 3.8–34.3 g) lower PW, with a weaker effect of SBP in the same direction (Fig. 6 and Supplementary Table 18). Previous MR analyses of fetal blood pressure effects on BW have been inconsistent—those that used similar methods[15,47] to the current analyses found no fetal effects, while others using transmitted and nontransmitted alleles[16,42] supported fetal genetic predisposition to higher SBP being causally related to lower BW. Our findings for PW are similar to the latter.

### PW and later neuropsychiatric traits

The role of placenta in neurodevelopment and later psychiatric disease in the offspring is a growing research field[6,48]. Using data from the Initiative for Integrative Psychiatric Research (iPSYCH) cohort

($n = 100,094$), we did not, however, find fetal polygenic scores for PW to be associated with risk of four neuropsychiatric diseases (Supplementary Note and Extended Data Figs. 9 and 10). Also, no significant genetic correlations were found between PW and neuropsychiatric diseases (Supplementary Table 19).

### Discussion

In this GWAS of PW (Fig. 7), we identify 40 independent association signals. These partially overlap with known BW loci, but 12 are related predominantly or only to PW, with connections to placental development and function. We observe a maternal POE near *KCNQ1*. Moreover, we find that fetal genetically mediated higher PW raises preeclampsia risk and shortens gestational duration, as well as demonstrates a role for fetal insulin in regulating placental growth.

There was a clear predominance of fetal effect signals among the genome-wide significant loci, and fetal SNP heritability was almost double that of maternal SNP heritability (Extended Data Fig. 2). A total of 26 signals were classified as fetal-only, and a further three as fetal and maternal. Four loci showed no fetal effect and represent indirect maternal genetic effects acting on PW via the intrauterine environment. For loci also known to be associated with BW, our mode of association results were in good agreement with a previous classification[16].

MoBa trio data revealed a pronounced POE signal near *KCNQ1* carried by the maternally transmitted allele, in agreement with previously observed silencing of the paternal alleles of *KCNQ1* and nearby *CDKN1C*. The association signal was classified as predominantly PW and showed weaker evidence of association with BW. A nearby SNP in low LD was strongly associated with BW in the most recent GWAS[16] and showed a similar association with PW in the MoBa data. These results suggest that there may be multiple imprinted underlying causal variants in the

region, perhaps with different modes of action during pregnancy. The *KCNQ1* finding implies effects on fetal and placental growth through at least two pathways[49]. Placenta cells have a unique epigenetic profile that regulates their transcription patterns, which may be associated with adverse pregnancy outcomes if disturbed. Beckwith–Wiedemann syndrome is an overgrowth syndrome with elevated PW and BW[49] where approximately 50% have lost maternal-specific methylation at the putative imprinting control region 2 (*ICR2*) within intron 10 of *KCNQ1* that is also the transcriptional start site of the regulatory long noncoding RNA *KCNQ1OT1* (ref. 50) controlling expression of *CDKN1C* and *KCNQ1*. The location of the lead signal, rs2237892, to this region suggests that it influences PW through an effect on methylation at *ICR2* in the placenta. Known associations between SNP rs2237892 and altered insulin secretion[51,52] support a role for fetal insulin in mediating the association with PW. Additionally, placental DNA methylation at *KCNQ1* has been associated with maternal insulin sensitivity[53], influencing maternal glucose availability across the placenta, and stimulating fetal insulin thus leading to fetal and placental growth.

Our results support a role for fetal genetic factors that predispose to a higher PW in raising preeclampsia risk. For every 1-s.d. higher genetically mediated PW, there were more than 1.5-fold higher odds of preeclampsia, a poorly understood complication occurring in 3–5% of pregnancies[4,54]. The causal relationship we observed appears specific to placental growth, with no similar effect of fetal genetic predisposition to higher BW. This finding may seem initially counter-intuitive, considering preeclampsia is often linked with fetal growth restriction. However, the relationship between preeclampsia and PW is not consistent—early onset preeclampsia (<34 weeks of gestation), which is likely caused by defective placentation, is frequently associated with fetal growth restriction and low PW[55,56], but the more common, 'late-onset' or 'term' preeclampsia (delivery in or after gestational week 37) is associated with normal placentation and has been associated observationally with both lower and higher PW[56]. Late-onset preeclampsia is thought to result largely from interactions between a maternal genetic susceptibility to cardiometabolic diseases and aging of the placenta[55]. Increased fetoplacental demands may result in uteroplacental mismatch and preeclampsia[4]. Our findings that fetal genetic factors predisposing to a larger placenta at term are associated with higher odds of preeclampsia are consistent with this model of late-onset preeclampsia. The available GWAS data on preeclampsia[38] used in our analysis was not stratified by gestational age at onset, so we recommend follow-up in stratified analyses to confirm that the association pertains to late-onset preeclampsia. However, the findings may open new opportunities for understanding the development of preeclampsia, its potential prevention and treatment.

Using PW to proxy placental growth enabled the large sample size of this study, leveraging routine records in birth registers, thereby yielding sufficient statistical power for discovery. One limitation, however, is that PW measured after delivery only crudely proxies placental growth and does not directly capture placental insults or other indications of placental dysfunction. Our study would therefore be complemented well by other approaches, for example, single-cell transcriptomics of placental cells early in pregnancy[19], characterization of placental mosaicism[57], sequencing of cell-free RNA transcripts of placental origin in maternal circulation during pregnancy[58] and RNA sequencing of placental samples without and with pregnancy complications[28,59]. Furthermore, while our results point to plausible candidate genes (for example, *FCGR2A*, *FCGR2B*, *TBX20*, *SERPINA1* and *SLC7A5*) and potential effects on placental development and morphology, and transport of antibodies and amino acids, follow-up studies are required to establish causal links and to characterize mechanisms related to placental growth.

Phenotype measurement and study heterogeneity are key considerations in the design of GWA meta-analyses[60]. Within a given cohort, phenotyping heterogeneity is likely to be below, as demonstrated in MoBa[61].

Across studies, however, variation in phenotyping procedures can occur, for example, regarding the trimming of membranes and clamping times. Within our study, we have addressed concerns regarding whether the placentas are trimmed or untrimmed and interstudy variability in placenta collection and measure by having the individual cohorts using *z* scores to standardize their PW. By using *z* scores, the measure of PW is in consideration to the mean and s.d. of each cohort, resulting in between-cohort equivalence of measure. The vast majority of our cohorts (85%) had gestational age measured through ultrasound. Considering that there is concordance between the gestational age and sex-adjusted and sex-only adjusted results, any heterogeneity of the gestational age assessment is estimated to have negligible influence. Finally, the effects of the abovementioned heterogeneity would mainly be study-specific as births and associated measures are subject to the obstetric policy. Within our meta-analyses, there was little evidence of between-study heterogeneity for lead variants (Supplementary Table 7 and Supplementary Fig. 6).

Our study complements large-scale GWA studies of BW. While genetic correlation and colocalization analyses revealed an overlap between signals associated with PW and BW, there were also notable differences emphasizing the complexity of common and distinct genetic regulation and physiological processes affecting placental and fetal growth[62]. Overall, our findings provide an improved understanding of the role of placenta in fetal growth and biological processes and complications such as gestational duration and preeclampsia. Future research may focus on additional proxies for placental function and environmental influences, as well as the role of placenta for later-life health outcomes.

## Online content

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

Robin N. Beaumont [1,95], Christopher Flatley [2,3,95], Marc Vaudel [4,95], Xiaoping Wu[5], Jing Chen [6,7], Gunn-Helen Moen [8,9,10,11,12], Line Skotte [5], Øyvind Helgeland [3,4], Pol Solé-Navais [2], Karina Banasik [13], Clara Albiñana [14], Justiina Ronkainen [15], João Fadista[5,16,17], Sara Elizabeth Stinson[18], Katerina Trajanoska [19,20], Carol A. Wang [21,22], David Westergaard [13,23,24], Sundararajan Srinivasan[25], Carlos Sánchez-Soriano[26], Jose Ramon Bilbao [27,28,29], Catherine Allard [30], Marika Groleau [31], Teemu Kuulasmaa [32], Daniel J. Leirer[1], Frédérique White [31], Pierre-Étienne Jacques[30,31], Haoxiang Cheng [33], Ke Hao [33], Ole A. Andreassen [34,35], Bjørn Olav Åsvold [10,36,37], Mustafa Atalay[32], Laxmi Bhatta[10], Luigi Bouchard[38,39], Ben Michael Brumpton [10,36,40], Søren Brunak [13], Jonas Bybjerg-Grauholm [41,42], Cathrine Ebbing[43,44], Paul Elliott [45], Line Engelbrechtsen[18,46], Christian Erikstrup [47,48], Marisa Estarlich [49,50,51], Stephen Franks[52], Romy Gaillard[53,54], Frank Geller [5], Jakob Grove [42,55,56,57], David M. Hougaard [41,42], Eero Kajantie [58,59,60], Camilla S. Morgen[18,61], Ellen A. Nohr[62], Mette Nyegaard [63], Colin N. A. Palmer [25], Ole Birger Pedersen [64,65], The Early Growth Genetics (EGG) Consortium*, Fernando Rivadeneira [19,53], Sylvain Sebert [15], Beverley M. Shields[1], Camilla Stoltenberg [66,67], Ida Surakka[68], Lise Wegner Thørner[69], Henrik Ullum[70], Marja Vaarasmaki[58,71], Bjarni J. Vilhjalmsson [14,57], Cristen J. Willer [68,72,73], Timo A. Lakka [32,74,75], Dorte Gybel-Brask [76], Mariona Bustamante[51,77,78], Torben Hansen [18], Ewan R. Pearson [25], Rebecca M. Reynolds[26], Sisse R. Ostrowski [65,69], Craig E. Pennell [21,22], Vincent W. V. Jaddoe [53,54], Janine F. Felix [53,54], Andrew T. Hattersley [1], Mads Melbye[10,65,79,80], Deborah A. Lawlor [11,81], Kristian Hveem[10,36], Thomas Werge [42,65,82,83], Henriette Svarre Nielsen [23,65], Per Magnus[84], David M. Evans [9,12,81], Bo Jacobsson [2,3], Marjo-Riitta Järvelin [45,85,86,87], Ge Zhang[7,88,89,90], Marie-France Hivert [91,92], Stefan Johansson [4,93,96] ✉, Rachel M. Freathy [1,81,96] ✉, Bjarke Feenstra [5,69,96] ✉ & Pål R. Njølstad [4,94,96] ✉

[1]Department of Clinical and Biomedical Sciences, Faculty of Health and Life Sciences, University of Exeter, Exeter, UK. [2]Department of Obstetrics and Gynecology, Institute of Clinical Sciences, Sahlgrenska Academy, University of Gothenburg, Gothenburg, Sweden. [3]Department of Genetics and Bioinformatics, Health Data and Digitalization, Norwegian Institute of Public Health, Oslo, Norway. [4]Mohn Center for Diabetes Precision Medicine, Department of Clinical Science, University of Bergen, Bergen, Norway. [5]Department of Epidemiology Research, Statens Serum Institut, Copenhagen, Denmark. [6]Division of Biomedical Informatics, Cincinnati Children's Hospital Medical Center, Cincinnati, OH, USA. [7]Department of Pediatrics, University of Cincinnati College of Medicine, Cincinnati, OH, USA. [8]Institute of Clinical Medicine, Faculty of Medicine, University of Oslo, Oslo, Norway. [9]Frazer Institute, The University of Queensland, Brisbane, Queensland, Australia. [10]K.G. Jebsen Center for Genetic Epidemiology, Department of Public Health and Nursing, NTNU, Norwegian University of Science and Technology, Trondheim, Norway. [11]Population Health Science, Bristol Medical School, University of Bristol, Bristol, UK. [12]Institute for Molecular Bioscience, The University of Queensland, Brisbane, Queensland, Australia. [13]Novo Nordisk Foundation Center for Protein Research, University of Copenhagen, Copenhagen, Denmark. [14]National Centre for Register-Based Research, Aarhus University, Aarhus, Denmark. [15]Research Unit of Population Health, University of Oulu, Oulu, Finland. [16]Department of Clinical Sciences, Lund University Diabetes Centre, Malmö, Sweden. [17]Institute for Molecular Medicine Finland (FIMM), University of Helsinki, Helsinki, Finland. [18]Novo Nordisk Foundation Center for Basic Metabolic Research, Faculty of Health and Medical Sciences, University of Copenhagen, Copenhagen, Denmark. [19]Department of Internal Medicine, Erasmus MC, University Medical Center Rotterdam, Rotterdam, The Netherlands. [20]Department of Human Genetics, McGill University, Montréal, Québec, Canada.

# Article

[21]School of Medicine and Public Health, College of Medicine, Public Health and Wellbeing, The University of Newcastle, Newcastle, New South Wales, Australia. [22]Hunter Medical Research Institute, New Lambton Heights, Newcastle, New South Wales, Australia. [23]Department of Obstetrics and Gynecology, Copenhagen University Hospital, Hvidovre, Denmark. [24]Methods and Analysis, Statistics Denmark, Copenhagen, Denmark. [25]Division of Population Health and Genomics, School of Medicine, University of Dundee, Dundee, UK. [26]Centre for Cardiovascular Science, The University of Edinburgh, Edinburgh, UK. [27]Department of Genetics, Physical Anthropology and Animal Physiology, University of the Basque Country (UPV/EHU), Leioa, Spain. [28]Biobizkaia Health Research Institute, Barakaldo, Spain. [29]Spanish Biomedical Research Center in Diabetes and Associated Metabolic Disorders (CIBERDEM), Barcelona, Spain. [30]Centre de recherche du Centre Hospitalier de l'Universite de Sherbrooke, Sherbrooke, Québec, Canada. [31]Département de Biologie, Faculté des Sciences, Université de Sherbrooke, Sherbrooke, Québec, Canada. [32]Institute of Biomedicine, School of Medicine, University of Eastern Finland, Kuopio Campus, Kuopio, Finland. [33]Icahn School of Medicine at Mount Sinai, New York City, NY, USA. [34]NORMENT Centre, Institute of Clinical Medicine, University of Oslo, Oslo, Norway. [35]Division of Mental Health and Addiction, Oslo University Hospital, Oslo, Norway. [36]HUNT Research Centre, Department of Public Health and Nursing, NTNU, Norwegian University of Science and Technology, Levanger, Norway. [37]Department of Endocrinology, Clinic of Medicine, St. Olavs Hospital, Trondheim University Hospital, Trondheim, Norway. [38]Department of Biochemistry and Functional Genomics, Faculty of Medicine and Health Sciences, Université de Sherbrooke, Sherbrooke, Québec, Canada. [39]Clinical Department of Laboratory Medicine, Centre intégré universitaire de santé et de services sociaux (CIUSSS) du Saguenay–Lac-St-Jean—Hôpital Universitaire de Chicoutimi, Saguenay, Québec, Canada. [40]Clinic of Medicine, St. Olavs Hospital, Trondheim University Hospital, Trondheim, Norway. [41]Department for Congenital Disorders, Statens Serum Institut, Copenhagen, Denmark. [42]iPSYCH, The Lundbeck Foundation Initiative for Integrative Psychiatric Research, Aarhus, Denmark. [43]Department of Clinical Science, University of Bergen, Bergen, Norway. [44]Department of Gynecology and Obstetrics, Haukeland University Hospital, Bergen, Norway. [45]Department of Epidemiology and Biostatistics, School of Public Health, Imperial College London, London, UK. [46]Department of Obstetrics and Gynecology, Herlev Hospital, Herlev, Denmark. [47]Department Clinical Immunology, Aarhus University Hospital, Aarhus, Denmark. [48]Department Clinical Medicine, Aarhus University, Aarhus, Denmark. [49]Faculty of Nursing and Chiropody, Universitat de València, C/Menendez Pelayo, Valencia, Spain. [50]Epidemiology and Environmental Health Joint Research Unit, Foundation for the Promotion of Health and Biomedical Research in the Valencian Region, FISABIO-Public Health, FISABIO-Universitat Jaume I-Universitat de València, Valencia, Spain. [51]Spanish Consortium for Research on Epidemiology and Public Health (CIBERESP), Madrid, Spain. [52]Institute of Reproductive and Developmental Biology, Imperial College London, London, UK. [53]The Generation R Study Group, Erasmus MC, University Medical Center Rotterdam, Rotterdam, The Netherlands. [54]Department of Pediatrics, Erasmus MC, University Medical Center Rotterdam, Rotterdam, The Netherlands. [55]Department of Biomedicine—Human Genetics and the iSEQ Center, Aarhus University, Aarhus, Denmark. [56]Center for Genomics and Personalized Medicine, Aarhus, Denmark. [57]Bioinformatics Research Centre, Aarhus University, Aarhus, Denmark. [58]Research Unit of Clinical Medicine, Medical Research Center, University of Oulu, Oulu, Finland. [59]Population Health Unit, Finnish Institute for Health and Welfare, Helsinki and Oulu, Oulu, Finland. [60]Department of Clinical and Molecular Medicine, Norwegian University of Science and Technology, Trondheim, Norway. [61]National Institute of Public Health, University of Southern Denmark, Copenhagen, Denmark. [62]Institute of Clinical research, University of Southern Denmark, Odense, Denmark. [63]Department of Health Science and Technology, Aalborg University, Aalborg, Denmark. [64]Department of Clinical Immunology, Zealand University Hospital, Køge, Denmark. [65]Department of Clinical Medicine, University of Copenhagen, Copenhagen, Denmark. [66]Norwegian Institute of Public Health, Oslo, Norway. [67]Department of Global Public Health and Primary Care, University of Bergen, Bergen, Norway. [68]Department of Internal Medicine, Division of Cardiology, University of Michigan, Ann Arbor, MI, USA. [69]Department of Clinical Immunology, Copenhagen University Hospital—Rigshospitalet, Copenhagen, Denmark. [70]Statens Serum Institut, Copenhagen, Denmark. [71]Department of Obstetrics and Gynaecology, Oulu University Hospital, Oulu, Finland. [72]Department of Biostatistics and Center for Statistical Genetics, University of Michigan, Ann Arbor, MI, USA. [73]Department of Human Genetics, University of Michigan, Ann Arbor, MI, USA. [74]Department of Clinical Physiology and Nuclear Medicine, Kuopio University Hospital, Kuopio, Finland. [75]Foundation for Research in Health Exercise and Nutrition, Kuopio Research Institute of Exercise Medicine, Kuopio, Finland. [76]Psychotherapeutic Outpatient Clinic, Mental Health Services, Capital Region, Copenhagen, Denmark. [77]ISGlobal, Institute for Global Health, Barcelona, Spain. [78]Universitat Pompeu Fabra (UPF), Barcelona, Spain. [79]Danish Cancer Institute, Copenhagen, Denmark. [80]Department of Genetics, Stanford University School of Medicine, Stanford, CA, USA. [81]Medical Research Council Integrative Epidemiology Unit, University of Bristol, Bristol, UK. [82]Institute of Biological Psychiatry, Mental Health Services, Copenhagen University Hospital, Copenhagen, Denmark. [83]Lundbeck Center for Geogenetics, GLOBE Institute, University of Copenhagen, Copenhagen, Denmark. [84]Centre for Fertility and Health, Norwegian Institute of Public Health, Oslo, Norway. [85]Center for Life Course Health Research, University of Oulu, Oulu, Finland. [86]MRC Centre for Environment and Health, School of Public Health, Imperial College London, London, UK. [87]Unit of Primary Health Care, Oulu University Hospital, OYS, Oulu, Finland. [88]Division of Human Genetics, Cincinnati Children's Hospital Medical Center, Cincinnati, OH, USA. [89]Center for Prevention of Preterm Birth, Perinatal Institute, Cincinnati Children's Hospital Medical Center, Cincinnati, OH, USA. [90]March of Dimes Prematurity Research Center Ohio Collaborative, Cincinnati Children's Hospital Medical Center, Cincinnati, OH, USA. [91]Department of Population Medicine, Harvard Medical School, Harvard Pilgrim Health Care Institute, Boston, MA, USA. [92]Diabetes Unit, Massachusetts General Hospital, Boston, MA, USA. [93]Department of Medical Genetics, Haukeland University Hospital, Bergen, Norway. [94]Children and Youth Clinic, Haukeland University Hospital, Bergen, Norway. [95]These authors contributed equally: Robin N. Beaumont, Christopher Flatley, Marc Vaudel. [96]These authors jointly supervised this work: Stefan Johansson, Rachel M. Freathy, Bjarke Feenstra, Pål R. Njølstad. *A list of authors and their affiliations appears at the end of the paper. ✉e-mail: stefan.johansson@uib.no; r.freathy@exeter.ac.uk; fee@ssi.dk; pal.njolstad@uib.no

## The Early Growth Genetics (EGG) Consortium

Robin N. Beaumont[1,95], Christopher Flatley[2,3,95], Marc Vaudel[4,95], Xiaoping Wu[5], Gunn-Helen Moen[8,9,10,11,12], Line Skotte[5], Øyvind Helgeland[3,4], Pol Solé-Navais[2], Justiina Ronkainen[15], João Fadista[5,16,17], Sara Elizabeth Stinson[18], Carol A. Wang[21,22], Jose Ramon Bilbao[27,28,29], Mustafa Atalay[32], Paul Elliott[45], Romy Gaillard[53,54], Frank Geller[5], Camilla S. Morgen[18,61], Ellen A. Nohr[62], Sylvain Sebert[15], Timo A. Lakka[32,74,75], Mariona Bustamante[51,77,78], Torben Hansen[18], Rebecca M. Reynolds[26], Craig E. Pennell[21,22], Vincent W. V. Jaddoe[53,54], Janine F. Felix[53,54], Andrew T. Hattersley[1], Mads Melbye[10,65,79,80], Deborah A. Lawlor[11,81], Per Magnus[84], David M. Evans[9,12,81], Bo Jacobsson[2,3], Marjo-Riitta Järvelin[45,85,86,87], Ge Zhang[7,88,89,90], Marie-France Hivert[91,92], Stefan Johansson[4,93,96], Rachel M. Freathy[1,81,96], Bjarke Feenstra[5,69,96] & Pål R. Njølstad[4,94,96]

A full list of members appears in the Supplementary Information.

## Methods

### Study cohorts, phenotype handling and ethical approval

Studies of individuals from European populations conducted GWA studies of PW as part of the Early Growth Genetics (EGG) Consortium. Where data were available, multiple births, congenital anomalies and babies born before 37 completed weeks of gestation or after 42 weeks and 6 d of gestation were excluded. Additionally, PW values <200 g or >1,500 g, or greater than 5 s.d. from the mean were excluded. The PW data were transformed into z scores for analysis—each individual z score was calculated using the within-cohort, unadjusted mean and s.d. of PW. Details for phenotype exclusions, data collection, sample size, mean PW and gestational age for each cohort can be found in Supplementary Tables 1, 3 and 5. Details of ethical approvals for the contributing studies are found in the Supplementary Information.

### Fetal, maternal and paternal meta-analyses

We performed separate GWA studies within each cohort to test associations between PW and fetal, maternal and paternal genotypes, as detailed in Eqs. (1)–(3).

$$\text{pw} \sim \text{child} + \text{sex} \, (+ \, \text{gestational age}) + \text{study specific covariates} \quad (1)$$

$$\text{pw} \sim \text{mother} + \text{sex} (+ \, \text{gestational age}) + \text{study specific covariates} \quad (2)$$

$$\text{pw} \sim \text{father} + \text{sex} \, (+ \, \text{gestational age}) + \text{study specific covariates} \quad (3)$$

where 'pw' refers to the standardized PW, and 'child', 'mother' and 'father' refer to the number of tested alleles (as genotype probabilities) for a given variant in the child, mother and father genomes, respectively.

Analyses were conducted twice, once adjusting for sex and gestational age and once adjusting for sex-only. Adjustment was also made for ancestry principal components, and the number of principal components included was determined on a per-cohort basis. Genotypes were imputed to the Haplotype Reference Consortium reference panel in most studies, with exceptions noted in Supplementary Tables 2, 4 and 6. Details of imputation and analysis for individual studies are shown in Supplementary Tables 2, 4 and 6. Association results from fetal, maternal and paternal GWA studies using both adjustment strategies were combined separately in fixed-effects meta-analyses implemented in METAL[63], resulting in six meta-analyses. Meta-analyses were performed independently by two analysts. SNPs were excluded if they were present in fewer than two studies or the number of individuals for the SNP was <5,000. Genome-wide significant loci were defined as regions with one or more SNP with $P < 5 \times 10^{-8}$, and these SNPs were defined as belonging to different loci if the distance between them was >500 kb. The lead SNP at each locus was the one with the smallest $P$ value. Secondary signals within each locus were identified using approximate conditional and joint multiple-SNP (COJO) analysis performed using GCTA–COJO[64]. Independent SNPs were defined as those with conditional $P < 5 \times 10^{-8}$. The LD reference panel was made up of 344,241 individuals from the UK Biobank defined as having British ancestry[65].

### gSEM

To calculate the SNP heritability of the fetal, maternal and paternal contributions, we used a framework within gSEM[21] developed by Moen et al.[22]. In short, the gSEM method[21] involves two stages. In the first stage, LD score regression methods using precomputed LD scores from a European population provided by the original developers of LD score regression[27,66] are applied to GWAS summary results statistics to estimate the genetic variance of each trait, and the genetic covariance between traits. In the second stage of gSEM, a user-defined SEM is fit to the genetic covariance matrix and parameters and their s.e. are estimated. The method is not restricted to trios and uses the full meta-analysis summary statistics.

The gSEM we used to partition genetic covariances into maternal and offspring components is displayed in Supplementary Fig. 7. Results from the three PW GWA studies (squares) were modeled in terms of latent maternal, paternal and offspring genetic variables (circles). The lower part of this model reflects simple biometrical genetics principles (that is, the fact that offspring and maternal genomes are correlated by 0.5) and consists of path coefficients fixed to the value one or one-half. The top half of the model consists of free parameters requiring estimation—three SNP heritabilities (one for each trait) and three genetic covariances between the variables, representing commonalities in genetic action across the fetal, maternal and paternal genomes[22].

It is important to realize that fitting a complicated SEM like the one in Supplementary Fig. 7 is necessary to obtain asymptotically unbiased estimates of SNP heritabilities and genetic correlations. The reason is that GWA studies of perinatal traits represent a complicated mixture of fetal, maternal and paternal genetic effects. Our SEM disentangles these effects from each contributing GWAS. In contrast, the model underlying LD score regression makes no allowance for this complication, and hence naïve use will lead to biased estimates of SNP heritability and genetic correlations containing an unknown mixture of fetal, maternal and paternal effects[22]. Summary results in statistics files from the GWA studies described above were combined using gSEM[21]. The software was set to not exclude insertions and deletions.

### Partitioning effects and allele transmission analysis

Partitioning fetal, maternal and paternal effects was performed from the summary statistics obtained from the GWAS (Eqs. (1)–(3)) using a WLM similar to that applied in ref. 15, extended to include paternal data and overlap in individuals between GWA studies. The estimates of partitioned fetal, maternal and paternal effects ($\eta_c$, $\eta_m$ and $\eta_f$, respectively) estimated from the GWAS estimates ($\beta_c$, $\beta_m$ and $\beta_f$, respectively) are

$$\eta_c = 2\beta_c - \beta_m - \beta_f \quad (4)$$

$$\eta_m = -\beta_c + \frac{3}{2}\beta_m + \frac{1}{2}\beta_f \quad (5)$$

$$\eta_f = -\beta_c + \frac{1}{2}\beta_m + \frac{3}{2}\beta_f. \quad (6)$$

To account for sample overlap between GWA studies, the covariance in estimates of the regression coefficients has the form

$$\sigma_{ij} = \frac{n_{sij}}{\sqrt{}} \quad (7)$$

where $n_{sij}$ is the number of individuals contributing to both analyses, $n_i$ is the number of individuals contributing to analysis $i$, $\rho_{ij}$ is the correlation between the estimates $\beta_i$ and $\beta_j$ in the overlapping samples and $\sigma_i$ is the s.e. of $\beta_i$. The term $\frac{n_s}{\sqrt{}}$ can be estimated as the intercept of a bivariate LD score regression[27]. The s.e. for partitioned effects are then estimated as

$$\sigma_{\eta c} = 4\sigma_c + \sigma_m + \sigma_f + 2\frac{n_{smf}}{\sqrt{}} \quad (8)$$

$$\sigma_{\eta m} = \frac{9}{4}\sigma_m + \sigma_c + \frac{1}{4}\sigma_f - \frac{n_{scf}}{\sqrt{}} \quad (9)$$

$$\sigma_{\eta f} = \frac{9}{4}\sigma_f + \sigma_c + \frac{1}{4}\sigma_m - \frac{n_{scm}}{\sqrt{}}. \quad (10)$$

$P$ values are calculated from WLM estimates using a z test, with test statistic

$$Z = \frac{\eta_i}{\sigma_i}. \quad (11)$$

To complement this analysis, we used the independent child–mother–father trios available in MoBa to perform conditional analyses where the association of fetal and parental genotypes with PW is conditioned against each other, as detailed in Eq. (12).

$$\text{Phenotype} \sim \text{child} + \text{mother} + \text{father} + \text{sex} + \text{gestational age} + \text{study specific covariates} \quad (12)$$

In this set of independent trios, using the phasing of the children's genotypes, we inferred the parent-of-origin of the genotyped alleles as done by Chen et al. [42]. We then studied the association of the PW with maternal and paternal transmitted and nontransmitted alleles, as detailed in Eq. (13).

$$\text{Phenotype} \sim \text{mnt} + \text{mt} + \text{pt} + \text{pnt} + \text{sex} + \text{gestational age} + \text{study specific covariates} \quad (13)$$

where 'mnt' and 'mt' refer to the maternal nontransmitted and transmitted alleles, respectively, and similarly, 'pt' and 'pnt' refer to the paternal transmitted and nontransmitted alleles, respectively.

To estimate the significance of effects mediated by the maternal transmitted only, and hence test for POEs, we conditioned it on the genotypes of the child, mother and father for the same variant, as detailed in Eq. (14).

$$\text{Phenotype} \sim \text{mt} + \text{child} + \text{mother} + \text{father} + \text{sex} + \text{gestational age} + \text{study specific covariates} \quad (14)$$

where variables are coded as in Eqs. (12) and (13). These analyses were conducted using z-scored PW and BW as phenotypes and hard-called genotypes. Each individual PW or BW z score was calculated using the within-study mean and s.d. Sex, gestational age and intercept were included in the model as well as study-specific covariates, that is, ten principal components and genotyping batches. The share of Mendelian errors was estimated using trios presenting at least a homozygous parent. If a variant presented more than 50% Mendelian errors, child alleles were swapped. All estimates are provided in Supplementary Table 11.

To classify SNPs as either fetal, maternal or paternal, we used similar criteria to those used previously for BW SNPs[15]. We classified SNPs into the following three categories: (1) fetal-only: the P value for the fetal estimate is lower than a Bonferroni corrected threshold (0.05/37 = 0.00132), and the 95% CI surrounding the estimate does not overlap the 95% CI for the maternal estimate; (2) maternal only: the P value for the maternal estimate is lower than a Bonferroni corrected threshold (0.00132), and the 95% CI surrounding the estimate does not overlap the 95% CI for the fetal estimate and (3) maternal and fetal: the P value for both the maternal and fetal association estimates were less than the Bonferroni corrected threshold (0.00132). If a SNP did not fit any of these criteria, it was marked as 'unclassified'.

### Assessing colocalization of association signals
To further supplement the aforementioned classifications, we performed colocalization analysis to determine overlap between GWAS signals from fetal, maternal and paternal meta-analyses using the R library 'coloc' version 5.1.0 (ref. 67) with the default prior probability for colocalization. Signals were defined as colocalising if the posterior probability of shared association signals (P4) was >0.8, distinct if the posterior probability of independent signals (P3) was >0.8 and undetermined if neither P3 or P4 was >0.8.

### LD score regression
To estimate the genetic correlation between PW and BW, LD score regression[27,66] was performed using summary statistics from gestational age and fetal sex-adjusted analyses and BW summary statistics taken from ref. 15. Estimates for the results from the meta-analyses and

WLM for both PW and BW were calculated for each of the fetal, maternal and paternal genomes.

### Testing association with BW
Colocalization analysis was again used to determine whether PW loci close to those previously identified in GWAS of BW represent the same association signal, using the same colocalization methods as above.

To classify SNPs as PW or BW, we calculated the 95% CI in s.d. units for the PW-lead SNP for both PW and BW. We then compared the 95% CIs; if these 95% CIs did not overlap, the SNP was classified as 'predominantly or only PW' unless either BW association (fetal or maternal) was in the opposite direction to the PW association and its associated P value was <0.05, in which case it was classified as PW and BW in opposite directions. SNPs whose 95% CIs for PW and BW overlapped were classified as 'PW and BW same direction'.

### Tissue-specific mRNA expression and scRNA data
We tested enrichment of expression of the PW-associated genes in specific tissues or cell types by comparing the ranks of gene expression across different tissue or cell types (Supplementary Fig. 8). This method involved the following two steps: first, the tissue or cell-type-specific expression levels from a reference expression dataset were rank normalized across all tissues or cell types for each gene; second, for a particular tissue or cell type, expression enrichment of the test genes were compared against all other genes by the Wilcoxon rank-sum test. We used 31 protein-coding genes close to the index SNPs of the 32 PW-associated loci identified in the fetal GWAS (first section of Table 1; the *LOC339593* locus was excluded due to the absence of nearby protein-coding genes). For the enrichment analysis in specific tissues, we used tissue-specific mRNA expression data from the Human Protein Atlas (RNA consensus tissue gene data)[32]. For the enrichment analysis in different cell types, we used the scRNA-seq data of about 70,000 single cells at the decidual–placental interface in early pregnancy[19]. Cell type-specific gene expression of each gene was evaluated by the percentage of cells with detectable scRNA reads. Significance levels for this enrichment analysis were Bonferroni corrected by number of tissues ($n = 61$; $P < 8.20 \times 10^{-4}$) or cell types ($n = 32$; $P < 1.56 \times 10^{-3}$).

### Identification of placental meQTL
Using the 41 SNPs identified in the discovery meta-analyses GWAS of PW, we conducted placental meQTL analysis in 395 participants with European ancestry from the Canadian cohort of the study Genetics of Glucose Regulation in Gestation and Growth (Gen3G)[29,30]. We measured DNA methylation using Illumina EPIC arrays v1.0 from samples collected on the fetal-facing side of the placenta. Methylation data were imported into R 4.1.0, and we performed quality control using the R package PACEanalysis v0.1.7 (refs. 68,69). Based on the results of the functions ExploratoryDataAnalysis and preprocessingofData, samples that failed as well as those with a sex or genetic mismatch, potential maternal contamination or low synciotrophoblast fraction (planet method) were removed, along with probes with nonsignificant detection ($P > 0.05$) for 5% or more of the samples[69–72]. Using the function preprocessingofData from the PACEanalysis package, we used minfi v1.38.0 (ref. 73) to correct for dye-bias and Noob[74] (from the package minfi) for background correction, followed by functional normalization[75]. We adjusted for probe-type bias using β-mixture quantile normalization[76], and we used the ComBat function from the package sva v3.40.0 to adjust for batch effect[77]. With the function detectionMask from PACEanalysis, we set the values as missing for probes with detection P value larger than 0.05, and with the function outlierprocess we windsorized outliers (1% extreme values). Finally, we removed probes annotated to sex-chromosomes, non-CpG probes, SNP-associated probes at the single base extension with a minor allele frequency larger than 5%, probes with an SNP at the target CpG with a minor allele frequency larger than 5%, cross-reactive probes previously identified[72]

and CpGs with low variance (variance <$1 \times 10^{-5}$, approximately 5% of the data). Whole-genome sequencing data were processed with the DNA-Seq v3.1.4 pipeline from GenPipes[78] based on BWA_mem and GATK best practices to identify high-quality SNPs on GRCh37 through joint genotyping over all samples. SNPs with a call rate above 20%, Hardy–Weinberg equilibrium $P$ value above $1 \times 10^{-6}$ and minor allele counts above ten were included.

meQTL were identified using $\beta$ values of 681,795 CpGs with TensorQTL v1.0.6 (ref. 79) in a $cis$-window of 0.5 Mb and the following covariates: four genotype principal components, fetal sex and cell type composition (estimates based on DNA methylation). To correct for the multiple CpGs tested genome-wide, an FDR threshold of ≤0.05 was applied to the $q$ values measured from the $P$ values extrapolated from the β distribution by TensorQTL.

## Mapping to pathways and traits

Variants classified as PW through the SNP classification of PW or BW were investigated further to identify relevant pathways and related biology. Systematic lookups using the OpenTargets Platform, The Human Protein Atlas and the International Mouse Phenotyping Consortium. Each variant was assessed for which gene it is functionally implicated in through information gained regarding the variant's involvement in molecular phenotype experiments, chromatin interaction experiments, in silico function predictions and distance between the variant and the gene's canonical transcription start site. The most likely candidate gene was then used to investigate the placenta-relevant cells that the gene is expressed in using information on the Open Targets Platform and The Human Protein Atlas websites. The International Mouse Phenotyping Consortium website was used to identify mouse models that found associations with placental pathology after the removal of the gene in single-gene knockout mice. Finally, further information was garnered through searching publications for which the candidate gene was specifically implicated in placental biology.

## Lookups of SNPs in GWAS of other phenotypes

We looked at associations between our independent PW-associated SNPs and other phenotypes within the phenoscanner[80]. We looked for associations with the SNP itself, SNPs in LD ($r^2 = 1$ and $r^2 \geq 0.8$). We additionally looked at associations between the independent PW SNPs in GWA studies for nausea and vomiting of pregnancy, hyperemesis gravidarum[37], preeclampsia[38], gestational duration[39], miscarriage (recurrent and spontaneous)[40] and ten cytokines assayed from neonatal blood spots[41] (Supplementary Table 17).

## MR analysis

We then performed two-sample MR analyses with own and offspring PW as outcomes. The exposures used included height, fasting glucose, disposition index of insulin secretion, insulin sensitivity, SBP and DBP using the same genetic instruments as the previous GWA studies of BW (Supplementary Table 18)[15]. We estimated both the effect of each maternal exposure on PW and the effect of the fetal genetic predisposition to each exposure on PW. For the latter, we made the assumption that SNP-exposure effects in the fetus would be the same as in the adult samples in which they were identified. Effect sizes were converted to grams using the value of 1 s.d. of PW of 132.5 g. We additionally used two-sample MR with own PW as the exposure, and gestational age, and preeclampsia as outcomes. The SNP-PW associations were taken from our GWAS of PW and the WLM-adjusted PW analysis. The SNP associations for all other traits were taken from external sources (Supplementary Table 17). To estimate the independent fetal associations for gestational age and preeclampsia, we first transformed the preeclampsia log ORs to the liability scale[81], then applied the WLM in the same way as for PW. The preeclampsia-independent fetal effects were then transformed to log ORs for analysis. A population prevalence for preeclampsia of 4.6% was used for the transformations[82]. We applied the inverse-variance weighted MR method, with MR-Egger[83], weighted median and penalized weighted median[84] acting as sensitivity analysis to test for robustness to MR assumptions.

## Polygenic scores

To investigate a possible association between PW and future risk of neuropsychiatric disease, we conducted analyses in the iPSYCH cohort. The iPSYCH cohort is a Danish population-based case–cohort sample including 141,215 individuals, of which 50,615 constitute a population-representative sample and the remainder are individuals with one or more neuropsychiatric disease diagnoses[85]. After restriction to individuals of European ancestries born at gestational ages between 37 and 42 weeks, 100,094 individuals remained for analysis, including 20,328 cases of attention-deficit/hyperactivity disorder (ADHD), 28,672 cases of affective disorder, 17,362 cases of autism spectrum disorder, 4,052 cases of schizophrenia and population-based sample of 32,995 individuals without any one of the four disorders. PW data from the Danish Medical Birth Register were available for 33,035 of these individuals (9,200 cases of ADHD, 2,946 cases of affective disorder, 9,678 cases of autism spectrum disorder, 239 cases of schizophrenia and 13,113 controls).

We redid the fetal meta-analysis for PW excluding the iPSYCH data, and based on the resulting summary statistics and a genetic correlation matrix created from the iPSYCH genotypes, we used LD-pred2-auto (ref. 86) to generate PGS for PW. Next, we regressed observed PW on ten fetal PW PGS quantiles, including sex, gestational age and ten principal components as covariates in the model. The linear regressions were done separately for the population controls and the four neuropsychiatric diseases. We also used logistic regression to assess the association between either (1) observed PW, or (2) fetal PGS of PW, and future risk of neuropsychiatric disease, including sex, gestational age and ten principal components as covariates in the model. In these analyses, the observed PW and the fetal PGS were first standardized to have mean 0 and variance 1.

## Statistics and reproducibility

No statistical method was used to predetermine the sample size. No data were excluded from the analyses. The experiments were not randomized. The investigators were not blinded to allocation during experiments and outcome assessment.

## Reporting summary

Further information on research design is available in the Nature Portfolio Reporting Summary linked to this article.

## Data availability

Individual cohorts contributing to the meta-analysis should be contacted directly as each cohort has different data access policies. GWAS summary statistics from this study are available via the EGG website (https://egg-consortium.org/placental-weight-2023.html, https://www.ebi.ac.uk/gwas/), as well as the GWAS catalog (https://www.ebi.ac.uk/gwas/, accession numbers GCST90275189, GCST90275190, GCST90275191, GCST90275192, GCST90275193, GCST90275194, GCST90275195, GCST90275196, GCST90275197, GCST90275198, GCST90275199). Access to personal-level information from Gen3G (including methylation array data) is subject to controlled access according to participants' consent concerning sharing of personal data. Request for conditions of access and for data access should be addressed to Center Hospitalier Universitaire de Sherbrooke institutional ethics committee: ethique.recherche.ciussse-chus@ssss.gouv.qc.ca.

## Code availability

Analysis code is available from https://github.com/EarlyGrowthGenetics/placental_weight_code (ref. 87).

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

## Acknowledgements

We are grateful to M. Efremova and S.A. Teichmann (Wellcome Sanger Institute, Cambridge, UK) who shared the maternal–fetal interface single-cell RNA data. Full acknowledgements and details of supporting grants are found in the Supplementary Information.

## Author contributions

The central analysis and manuscript drafting team included R.N.B., C.F., M. Vaudel, X.W., L.S., G.-H.M., D.M.E., B.J., M.-R.J., G.Z., M.-F.H, S.J., R.M.F., B.F., P.R.N. Sample collection, genotyping and/or phenotyping were performed and/or directed by C.F., M. Vaudel, G.-H.M., Ø.H., P.S.-N., K.B., D.W., J.R.B., H.C., K. Hao, O.A.A., B.O.Å., M.A., L. Bhatta, L. Bouchard, B.M.B, S.B., J.B.-G., P.E., L.E., C. Erikstrup, M.E., S.F., R.G., F.G.E., J.G., D.M.H., E.K., C.S.M., E.A.N., M.N., C.N.A.P., O.B.P., F.R., B.M.S., C.S., I.S., L.W.T., H.U., M. Vaarasmaki, B.J.V., C.J.W., T.A.L., D.G.-B., M.B., T.H., E.R.P., R.M.R., S.R.O., C.E.P., V.W.V.J., J.F.F., A.T.H., M.M., D.A.L., K. Hveem, T.W., H.S.N., P.M., B.J., M.-R.J., M.-F.H., S.J., R.M.F., B.F., P.R.N. Statistical analysis was performed by R.N.B., C.F., M. Vaudel, X.W., J.C., G.-H.M., L.S., C. Albiñana, J.R., J.F., S.E.S., K.T., C.A.W., S. Srinivasan, C.S.S., J.R.B., C. Allard, M.G., T.K., D.J.L., F.W., H.C., K. Hao, M.A., F.G.E., J.G., B.J.V., T.A.L., M.B., J.F.F., D.M.E., G.Z., B.F. Data was interpreted by R.N.B., C.F., M. Vaudel, X.W., L.S., G.-H.M., J.C., Ø.H., P.S.-N., C. Albiñana, J.R., K.T., S. Srinivasan, C. Allard, M.G., F.W., P.-É.J., B.O.Å., L. Bhatta, B.M.B., C. Ebbing, P.E., S.F., J.G., E.K., C.N.A.P., F.R., S. Sebert, M. Vaarasmaki, C.J.W., E.R.P., R.M.R., V.W.V.J., J.F.F., D.A.L., K. Hveem, D.M.E., B.J., M.-R.J., G.Z., M.-F.H., S.J., R.M.F., B.F., P.R.N. All authors contributed to the final version of the manuscript.

## Competing interests

O.A.A. is a consultant to Cortechs.ai and has received speaker's fees from Lundbeck, Janssen and Sunovion. B.J.V. is on Allelica's scientific advisory board. C.J.W.'s spouse works for Regeneron Pharmaceuticals. D.A.L. has received support from Roche Diagnostics and Medtronic for research unrelated to that presented here. H.S.N. has received speaker's fees from Ferring Pharmaceuticals, Merck A/S, Astra Zeneca, Cook Medical and IBSA Nordic. S.B. has ownerships in Intomics A/S, Hoba Therapeutics Aps, Novo Nordisk A/S, Lundbeck A/S, ALK-Abelló A/S and managing board memberships in Proscion A/S and Intomics A/S. The remaining authors declare no competing interests.

## Additional information

**Extended data** is available for this paper at https://doi.org/10.1038/s41588-023-01520-w.

**Correspondence and requests for materials** should be addressed to Stefan Johansson, Rachel M. Freathy, Bjarke Feenstra or Pål R. Njølstad.

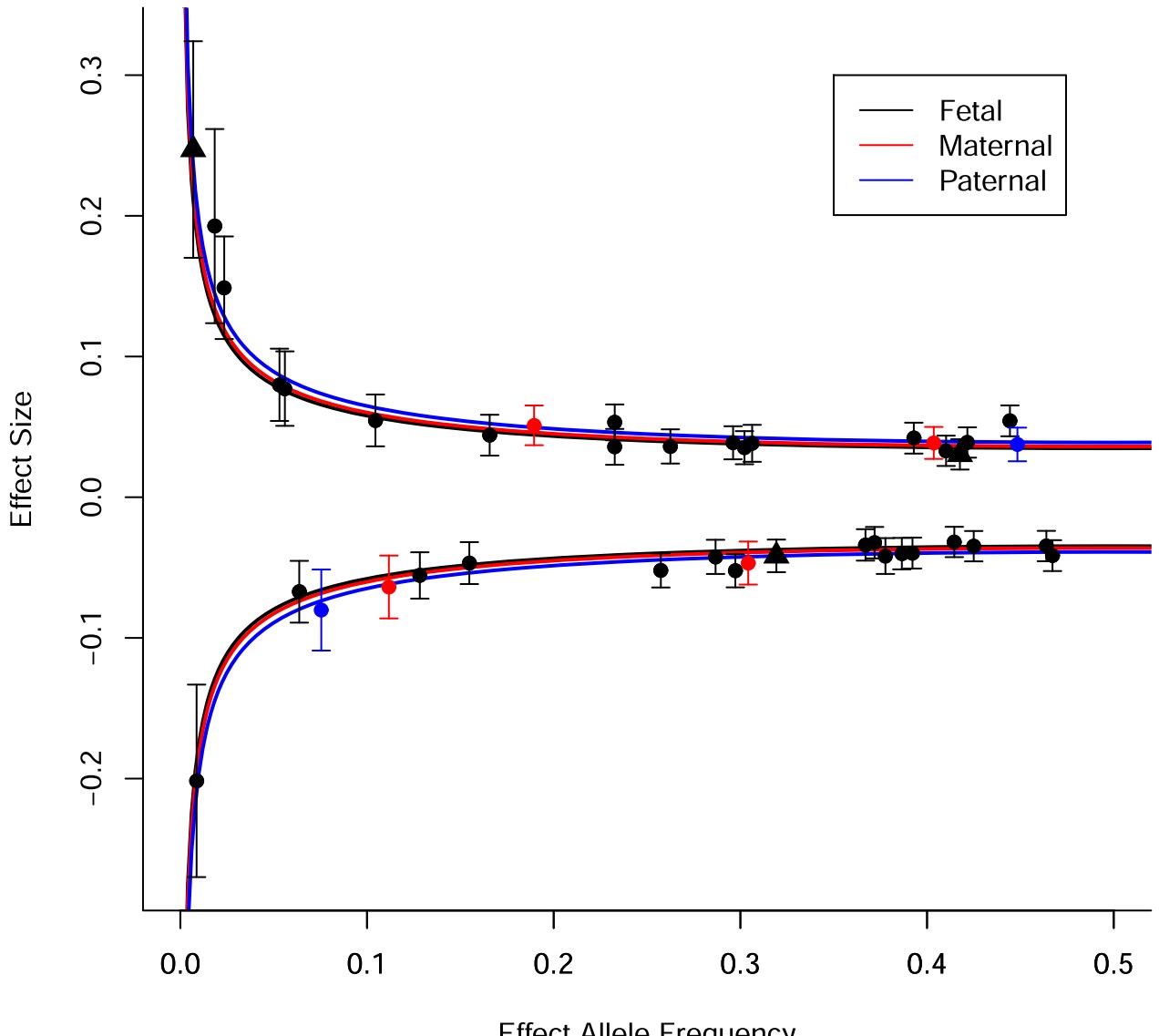

**Extended Data Fig. 1 | Effect sizes and minor allele frequencies for placental weight-associated lead SNPs.** Variants identified in the fetal, maternal and paternal analyses are shown in black, red, and blue, respectively. The lines indicate effect sizes needed to have 80% power to detect variants at genome-wide significance with the sample sizes of the fetal, maternal, and paternal analyses. Circles indicate main signals and triangles indicate secondary signals (fetal n = 65,405; maternal n = 61,228; paternal n = 52,392). Error bars represent 95% confidence intervals.

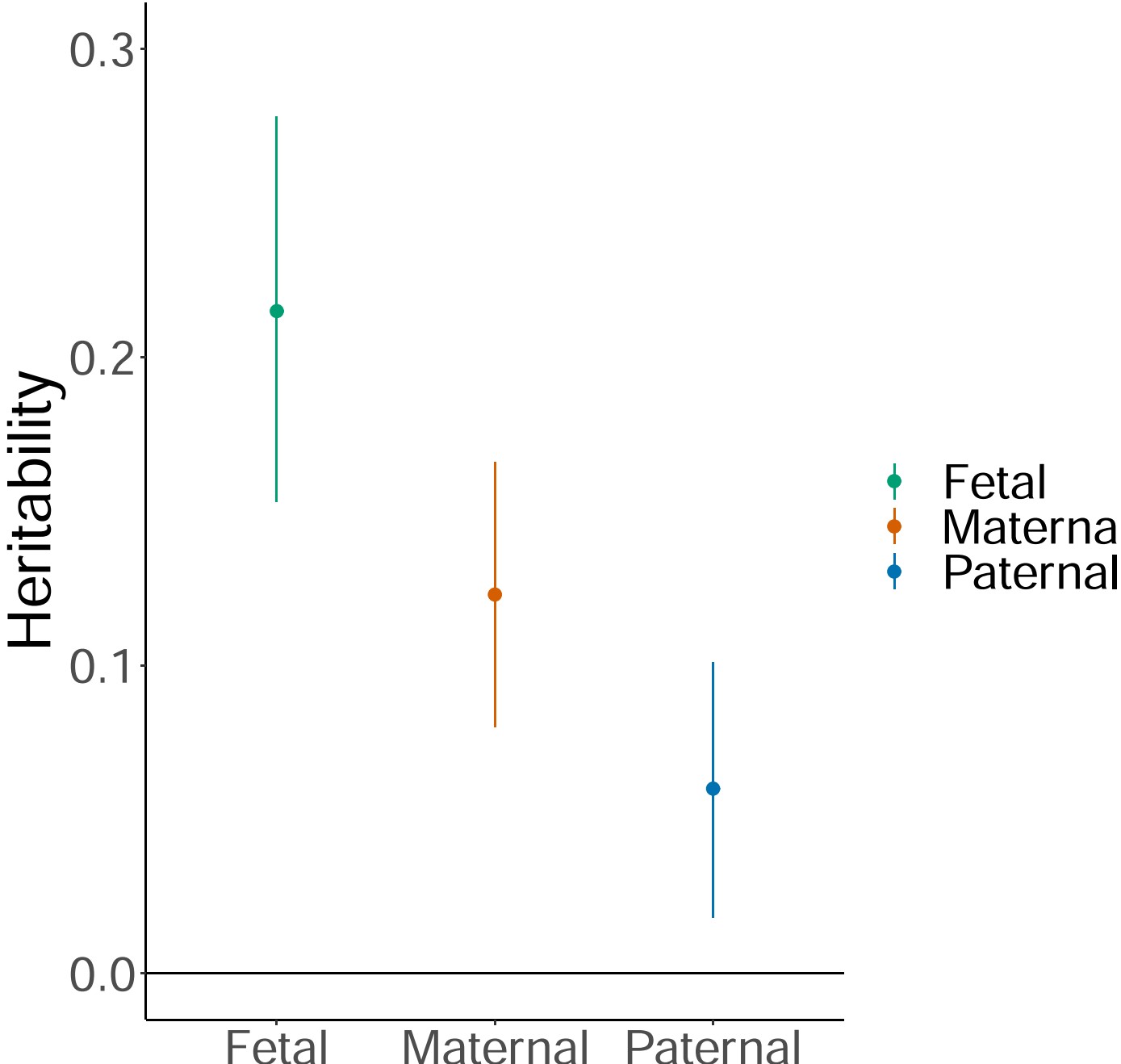

**Extended Data Fig. 2 | Heritability estimates for placental weight from genomic SEM analysis.** Scatter plot showing SNP heritability estimates ($h^2$) for fetal, maternal and paternal genomes estimated using genomic SEM (fetal n = 65,405; maternal n = 61,228; paternal n = 52,392). Error bars represent 95% confidence intervals.

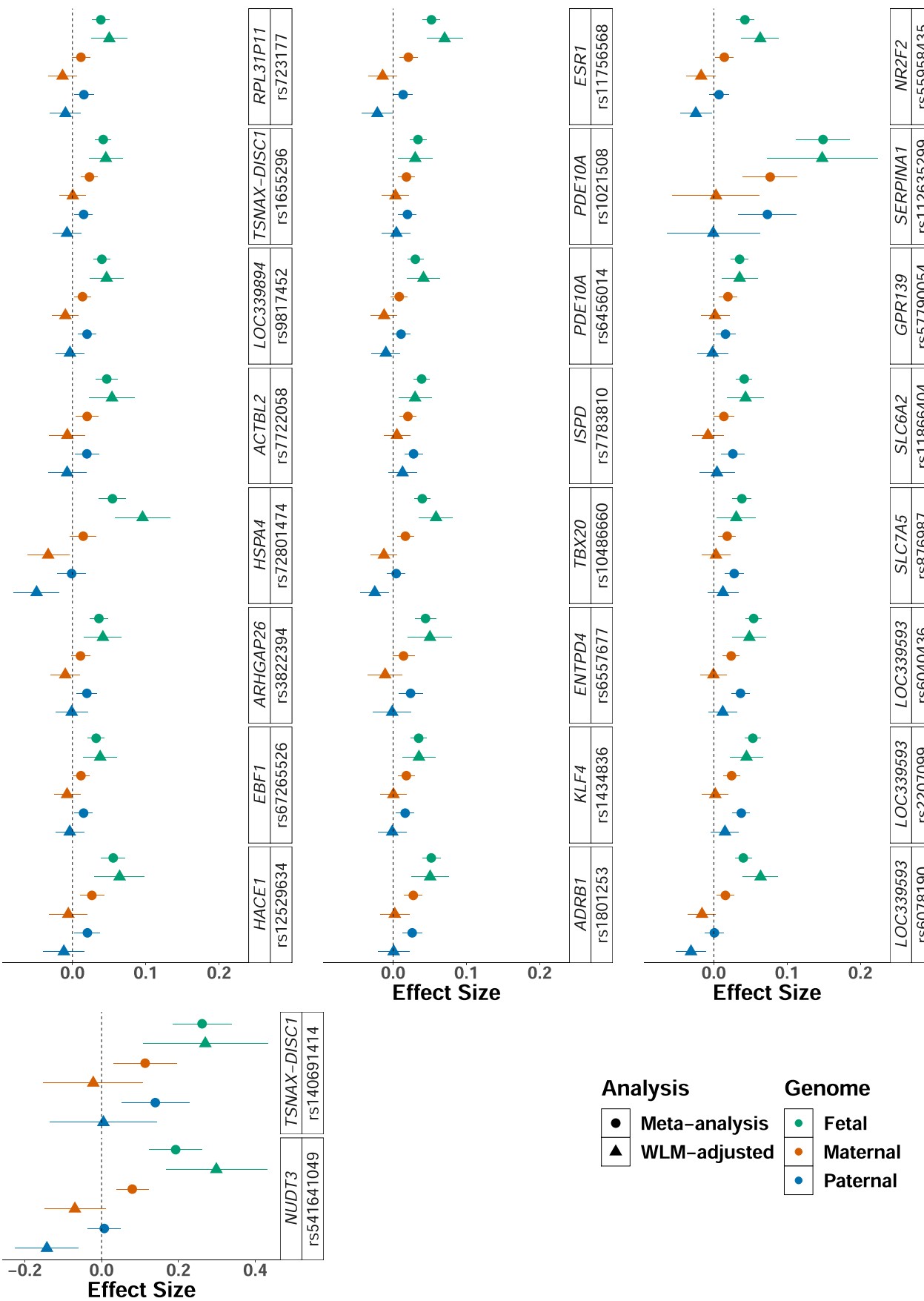

**Extended Data Fig. 3 | Fetal classified loci: effects from meta-analysis and weighted linear model for each genome.** Shown are the variants which were classified as having a fetal effect. Estimates are provided for fetal, maternal, and paternal effects for the meta-analysis results and after weighted linear model adjustment (fetal n = 65,405; maternal n = 61,228; paternal n = 52,392). Circles and triangles are association estimates, and error bars represent 95% confidence intervals. Abbreviation: WLM, weighted linear model. *Note different scale on x-axis for rs140691414 - *TSNAX-DISC1* & rs541541049 *NUDT3*.

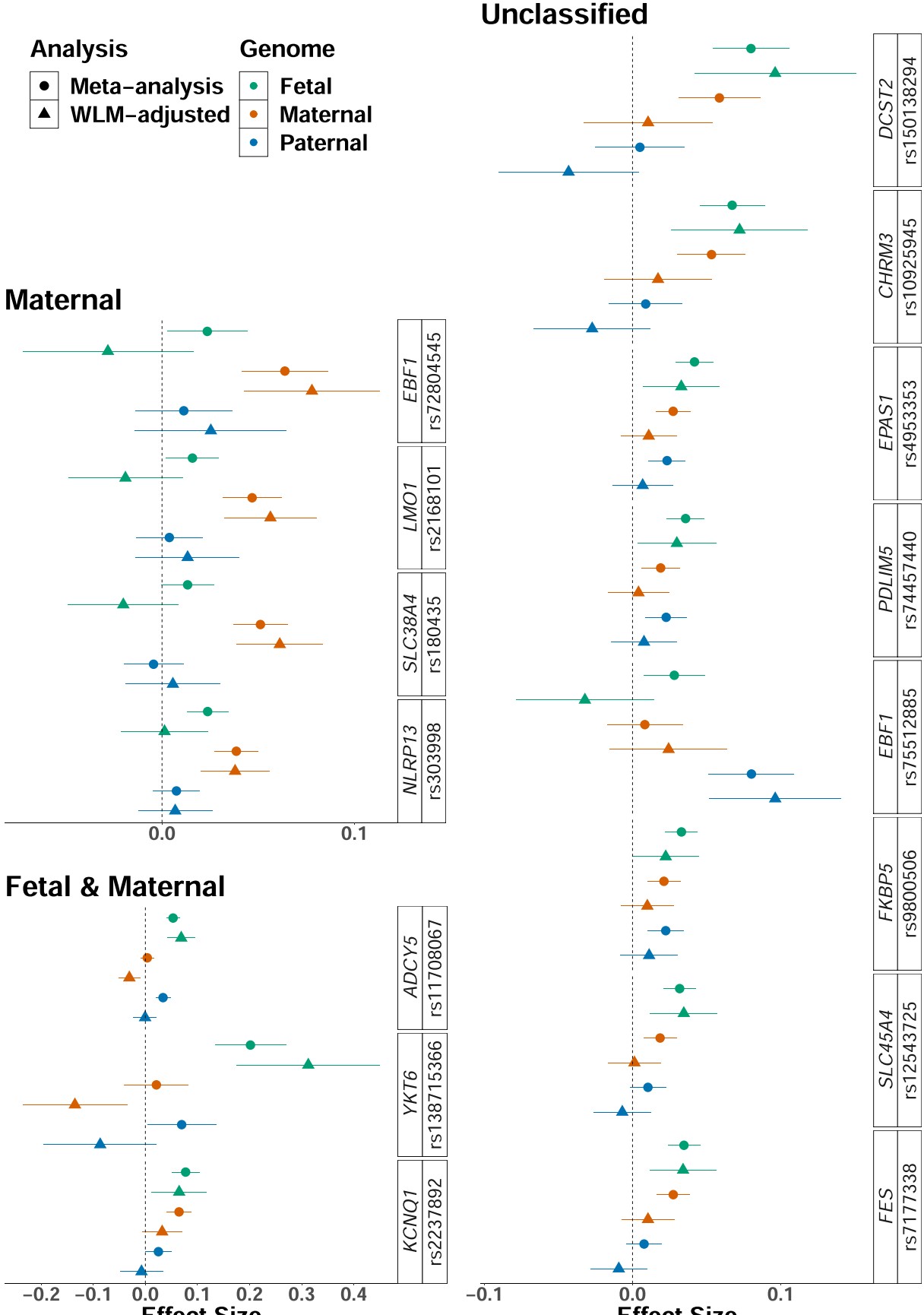

**Extended Data Fig. 4 | Classifications of remaining loci and effects from meta analysis and weighted linear model for each genome.** Shown are the remaining variants. Estimates are provided for fetal, maternal, and paternal effects for the meta-analysis results and after weighted linear model adjustment (fetal n = 65,405; maternal n = 61,228; paternal n = 52,392). Circles and triangles are association estimates, and error bars represent 95% confidence intervals. Abbreviation: WLM, weighted linear model.

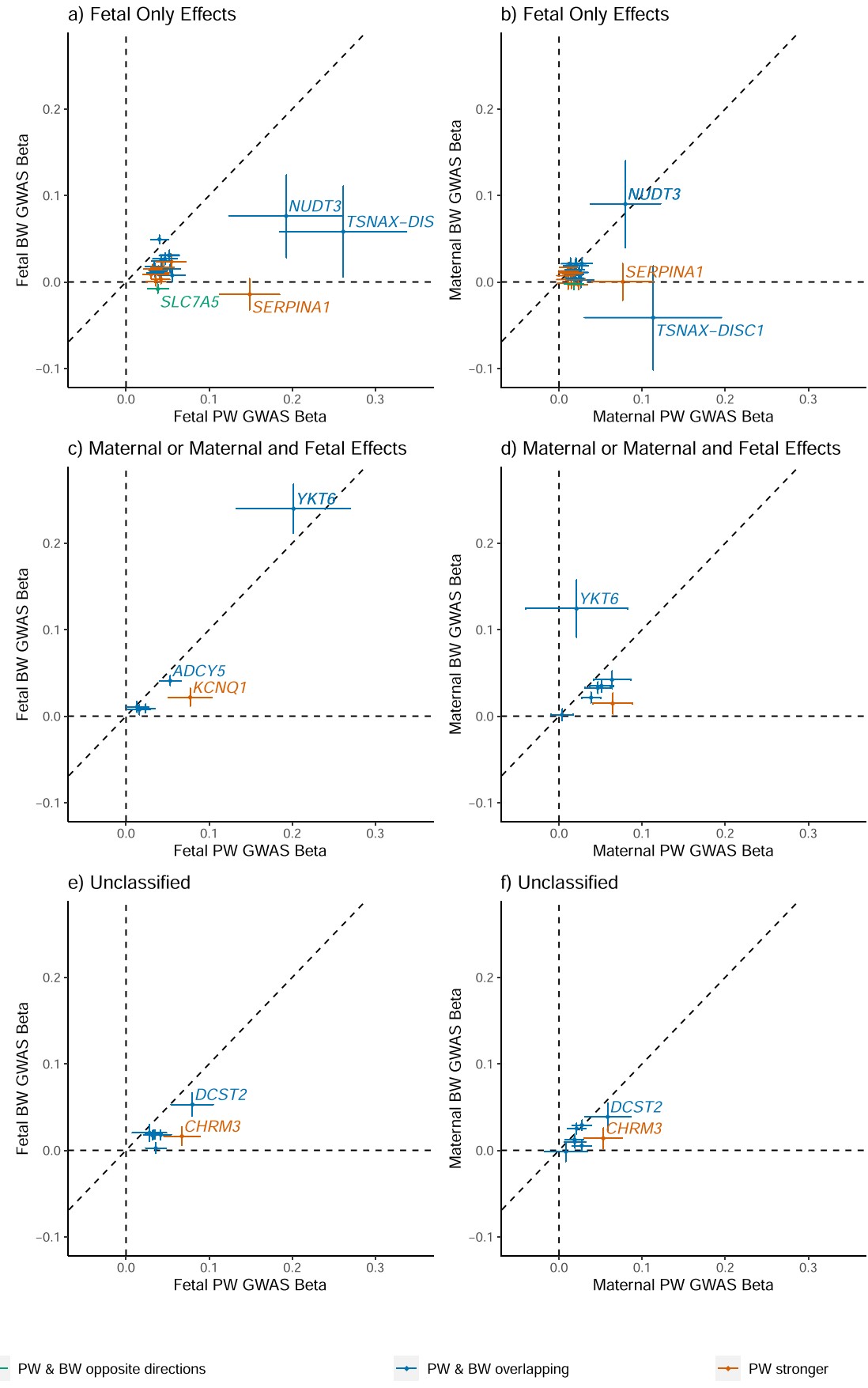

**Extended Data Fig. 5 | See next page for caption.**

**Extended Data Fig. 5 | Scatter plots comparing effect sizes from placental weight and birth weight GWAS for placental weight SNPs. a–e**, Scatter plots comparing effect size estimates and 95% confidence intervals from the placental weight GWAS (n = 65,405) with those from the birth weight GWAS (n = 321,223)[2]. Panels **a** and **b** show only SNPs classified as having fetal only effects, panels **c** and **d** show SNPs with maternal only or maternal and fetal effect, and panels **e** and **f** show unclassified SNPs. Panels **a**, **c** and **e** show fetal PW and BW betas, and panels **b**, **d** and **f** show maternal PW and BW betas. The left column shows fetal genome associations, and the right shows maternal. The top row shows SNPs classified as fetal only effects on PW (Supplementary Table 7). The middle row shows SNPs classified as maternal, or maternal and fetal, and the bottom row shows unclassified SNPs. Colors indicate classifications, which are given in a key below the figure. Abbreviations: BW, birth weight; GWAS, genome-wide association study; PW, placental weight. Error bars represent 95% confidence intervals.

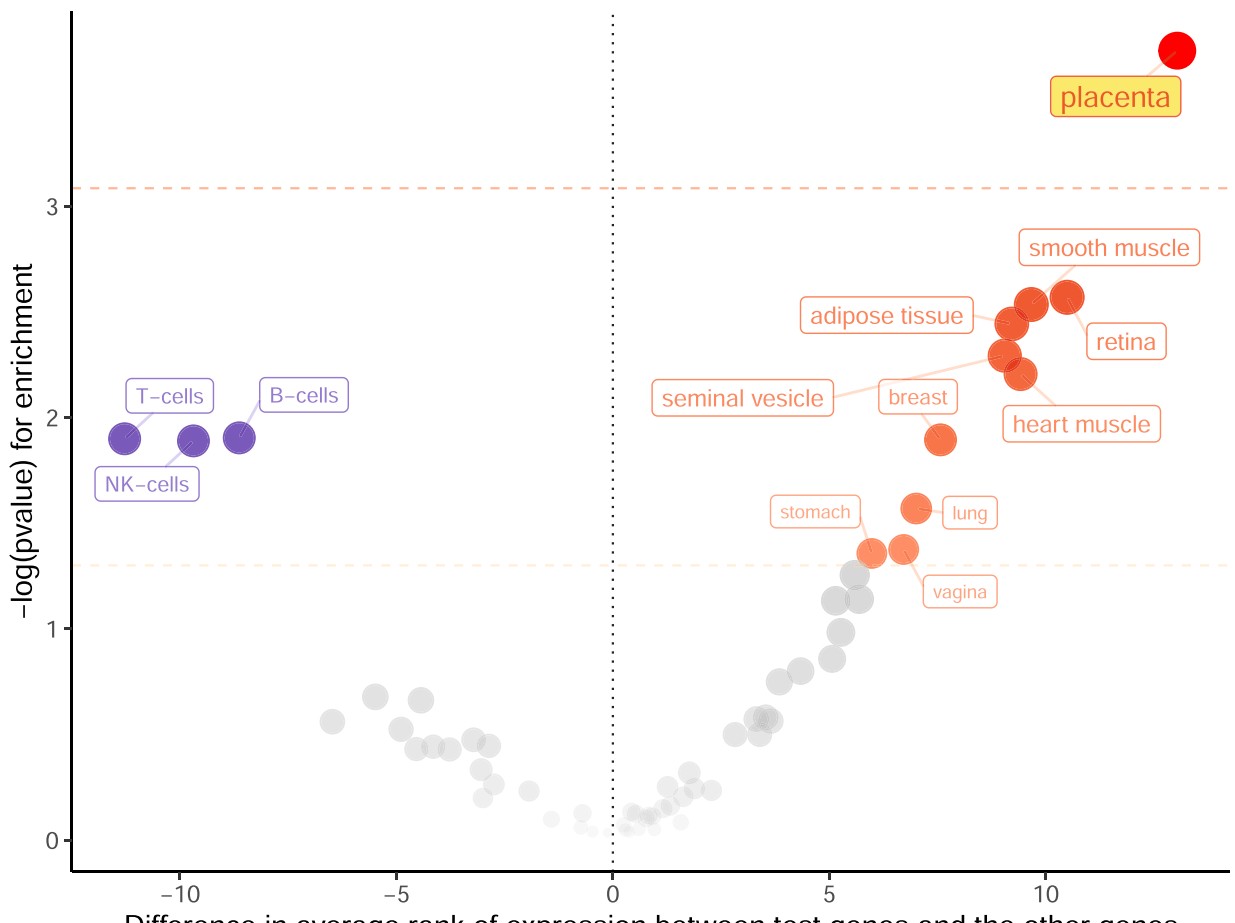

**Extended Data Fig. 6 | Tissue enrichment by mRNA data.** Plot illustrating the enrichment or depletion of RNA expression of the 31 placental weight-associated protein-coding genes identified in the fetal GWAS, in 61 different tissues. Each dot represents a specific tissue and plots the difference in average rank of expression levels between the 31 placental weight-associated genes and all the other genes (x-axis) with associated -log($P$ value) based on the Wilcoxon rank-sum test (y-axis). The size of the points is inversely proportional to the log $P$ value. The two dashed horizontal lines represent significance levels with (red) or without (orange) Bonferroni correction ($n = 61$). Tissues with nominally significant ($P$ value < 0.05) higher or lower expression of the test genes are plotted and labeled as red or blue dots, respectively. Tissues with Bonferroni corrected significance are highlighted by labels with yellow background.

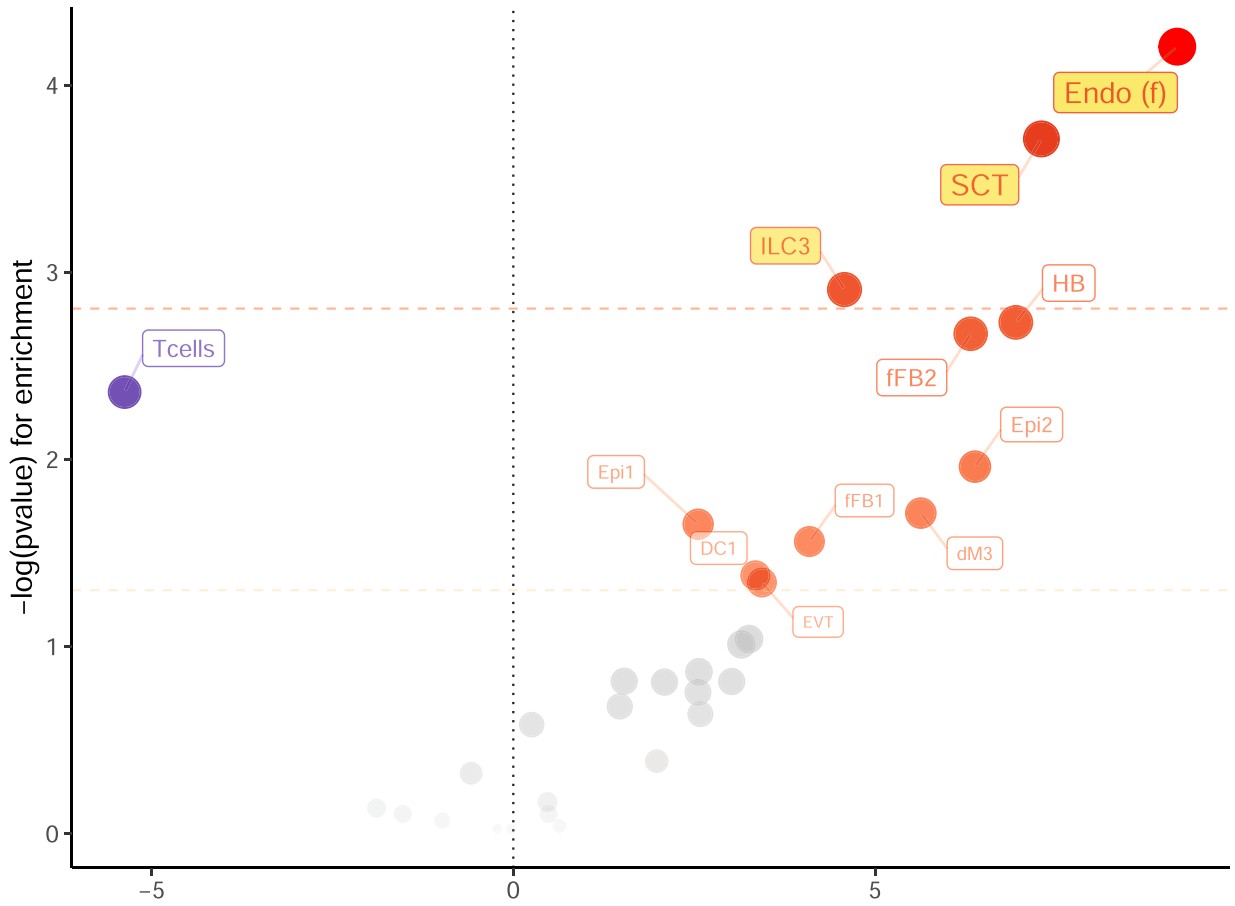

**Extended Data Fig. 7 | Cell-type enrichment by scRNA-seq data.** Plot illustrating the enrichment or depletion of RNA expression of the 31 nearest protein-coding genes at placental weight loci identified in the fetal GWAS in 32 different cell types at the early maternal-fetal interface. Each dot represents a specific cell-type and plots the difference in average rank of expression between the 31 placental genes and all the other genes (x-axis) with associated -log($P$ value) based on the Wilcoxon rank-sum test (y-axis). The size of the points is inversely proportional to the log $P$ value. The two dashed horizontal lines represent significance levels with (red) or without (orange) after Bonferroni correction ($n$ = 32). Cell types with nominally significant ($P$ value < 0.05) higher or lower expression of the test genes are plotted and labeled as red or blue dots, respectively. Cell types with Bonferroni corrected significance are highlighted by labels with yellow background. Abbreviations of cell types with nominally significant difference in expression: Endo (f), endothelial cells (fetal); SCT, syncytiotrophoblast (fetal); ILC, innate lymphocyte cells (maternal); HB, Hofbauer cells (fetal); fFB1 and fFB2, fibroblasts (fetal); Epi1 and Epi2, epithelial glandular cells (unassigned or maternal); dM3, Maternal macrophages (maternal cell in placenta); EVT, extravillous trophoblast; DC1, dendritic cells; T cells, T cells (maternal or fetal).

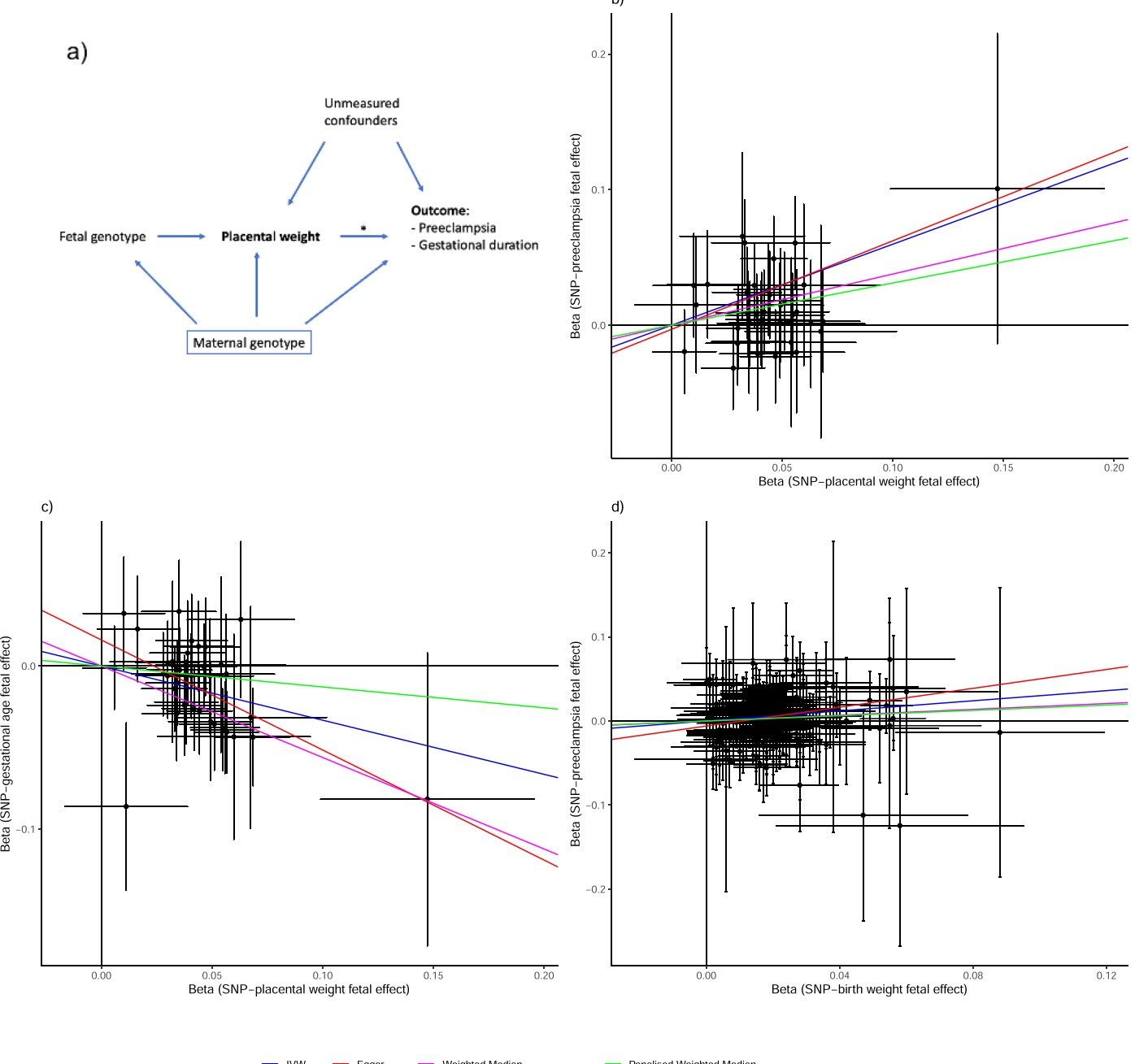

IVW — Egger — Weighted Median — Penalised Weighted Median

**Extended Data Fig. 8 | Mendelian randomization analyses. a**, Diagram illustrating the Mendelian randomization analyses used to test for a causal relationship (*) between higher placental weight (exposure; a proxy for faster placental growth) and either preeclampsia, or gestational duration (outcomes). Key assumptions: (i) fetal genotype (genetic instrumental variable) is robustly related to placental weight, (ii) potential confounders of the causal relationship of interest are not associated with the genetic instrumental variable, and (iii) the genetic instrumental variable is only related to the outcome via its effect on the exposure (placental weight), not through any other pathway. Since maternal genotype is correlated with fetal genotype and may additionally influence placental weight and the outcome variables, it is a potential confounder and should be adjusted for in the analyses (indicated by the box around it).

We were able to adjust for maternal genotype using weighted linear model (WLM) estimates of fetal genetic effects on placental weight and on preeclampsia and gestational duration, since both maternal and fetal GWAS summary statistics are available for those outcomes. To check for deviation from assumption (iii) above, we used the MR Egger, weighted median and penalized weighted median sensitivity analyses. **b**–**d**, Results of two-sample Mendelian randomization analyses testing the effect of (**b**) higher placental weight (n = 65,405) using fetal genetic instruments on preeclampsia (n = 167,234), (**c**) higher placental weight using fetal genetic instruments on gestational duration (n = 43,568), and (**d**) higher birth weight (n = 321,223) using fetal genetic instruments on preeclampsia. Points represent SNP effect estimates and error bars show 95% confidence intervals.

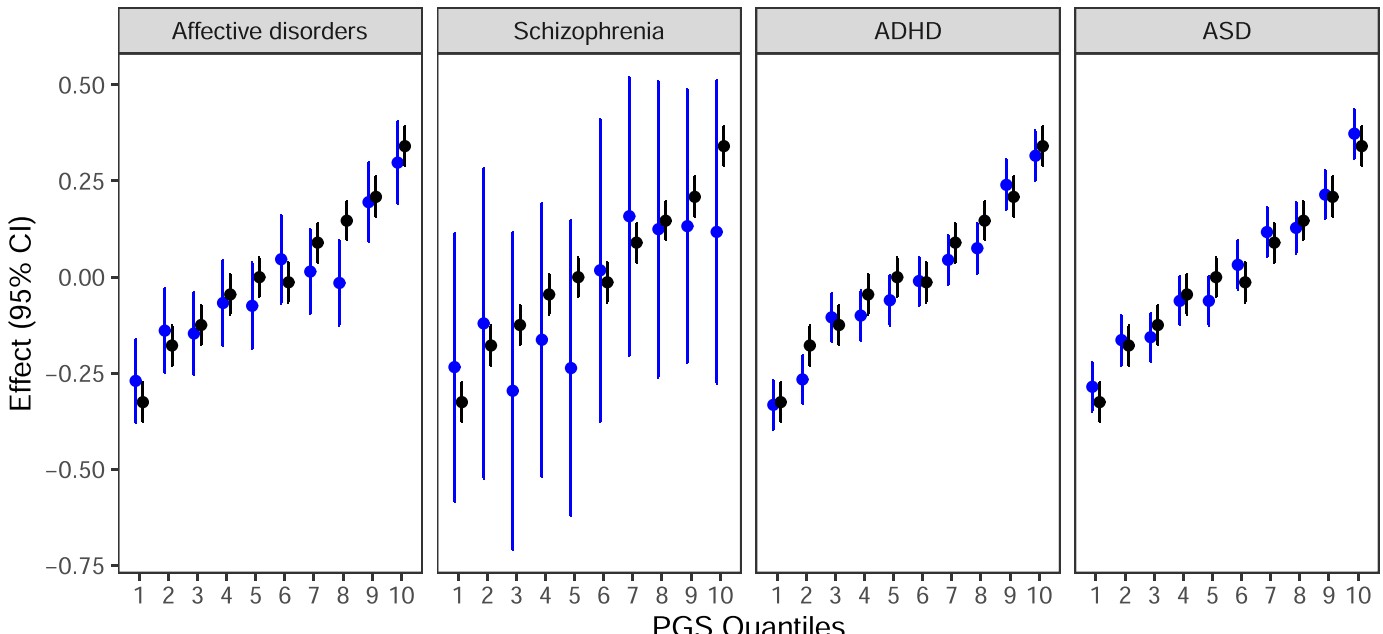

**Extended Data Fig. 9 | Polygenic score analyses.** The panels show associations between quantiles of fetal polygenic scores for placental weight and standardized observed placental weight in the iPSYCH cohort (n = 33,035). Points represent association effect estimates and error bars show 95% confidence intervals. Black dots show associations for the population representative sample used as controls in iPSYCH, and blue dots show associations for cases of four different neuropsychiatric diseases. Abbreviations: CI, confidence interval; ADHD, attention deficit/hyperactivity disorder; ASD, autism spectrum disorder; PGS, polygenic score.

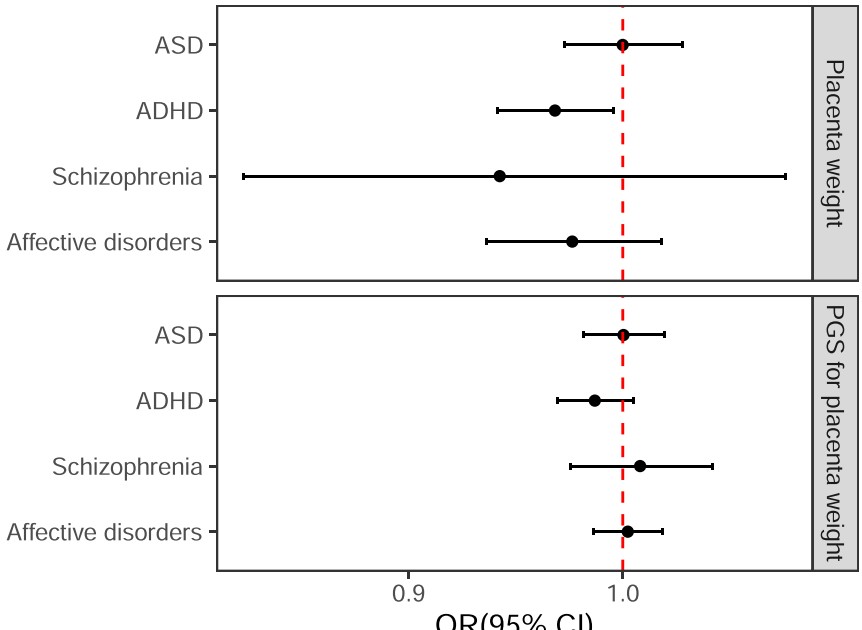

**Extended Data Fig. 10 | Analyses in the iPSYCH cohort of placental weight and risk of neuropsychiatric diseases.** The figure shows odds ratios (ORs) from logistic regressions of four neuropsychiatric diseases on standardized observed placental weight (upper panel) and fetal polygenic score of placental weight (lower panel). The ORs correspond to a change of one standard deviation in standardized observed placental weight or PGS for placental weight. Abbreviations: CI, confidence interval; ADHD, attention deficit/hyperactivity disorder; ASD, autism spectrum disorder; OR, odds ratio; PGS, polygenic score.

# Reporting Summary

Nature Research wishes to improve the reproducibility of the work that we publish. This form provides structure for consistency and transparency in reporting. For further information on Nature Research policies, see our Editorial Policies and the Editorial Policy Checklist.

## Statistics

For all statistical analyses, confirm that the following items are present in the figure legend, table legend, main text, or Methods section.

| n/a | Confirmed | |
|---|---|---|
| ☐ | ☒ | The exact sample size (*n*) for each experimental group/condition, given as a discrete number and unit of measurement |
| ☐ | ☒ | A statement on whether measurements were taken from distinct samples or whether the same sample was measured repeatedly |
| ☐ | ☒ | The statistical test(s) used AND whether they are one- or two-sided *Only common tests should be described solely by name; describe more complex techniques in the Methods section.* |
| ☐ | ☒ | A description of all covariates tested |
| ☐ | ☒ | A description of any assumptions or corrections, such as tests of normality and adjustment for multiple comparisons |
| ☐ | ☒ | A full description of the statistical parameters including central tendency (e.g. means) or other basic estimates (e.g. regression coefficient) AND variation (e.g. standard deviation) or associated estimates of uncertainty (e.g. confidence intervals) |
| ☐ | ☒ | For null hypothesis testing, the test statistic (e.g. *F*, *t*, *r*) with confidence intervals, effect sizes, degrees of freedom and *P* value noted *Give P values as exact values whenever suitable.* |
| ☐ | ☒ | For Bayesian analysis, information on the choice of priors and Markov chain Monte Carlo settings |
| ☐ | ☒ | For hierarchical and complex designs, identification of the appropriate level for tests and full reporting of outcomes |
| ☐ | ☒ | Estimates of effect sizes (e.g. Cohen's *d*, Pearson's *r*), indicating how they were calculated |

*Our web collection on statistics for biologists contains articles on many of the points above.*

## Software and code

Policy information about availability of computer code

| Data collection | Data collection was conducted by the cohorts participating in the study, software and code used is documented in the cohort study publications. |
|---|---|
| Data analysis | The software and version used for the GWAS in the different cohorts is listed in Supplementary Tables 2, 4, and 6. The code used to conduct the follow-up analyses is available at github.com/EarlyGrowthGenetics/placental_weight_code Briefly: <br>- Meta-analysis was conducted using Metal (version dated 2011-03-25). <br>- Conditional and joint analysis was conducted using GCTA-COJO v1.26.0 <br>- Colocation analyses were conducted using the R v4.0.0 package coloc v4.0-4 <br>- Structural equation modeling was conducted using genomic SEM v0.0.5 <br>- Trio and transmission analyses were conducted using WLM (script in repository) for the meta-analysis results and using TrioGen (https://github.com/mvaudel/trioGen v. 0.5.0) <br>- Placental meQTLs were identified using TensorQTL <br>- Polygenic scores were built using LD-pred2 <br>- Phasing was conducted using SHAPEIT, SHAPEIT2, SHAPEIT3, PBWT, Eagle v2.3, Eagle v2.4, or IMPUTE2 <br>- Imputation was conducted using IMPUTE2, IMPUTE3, Minimac3, Minimac4, PBWT, or the Sanger imputation server <br>- GWAS lookups were preformed using Phenoscanner v1.0 <br><br>Custom analysis code is available in https://github.com/EarlyGrowthGenetics/placental_weight_code |

For manuscripts utilizing custom algorithms or software that are central to the research but not yet described in published literature, software must be made available to editors and reviewers. We strongly encourage code deposition in a community repository (e.g. GitHub). See the Nature Research guidelines for submitting code & software for further information.

## Data

Policy information about availability of data

All manuscripts must include a data availability statement. This statement should provide the following information, where applicable:

- Accession codes, unique identifiers, or web links for publicly available datasets
- A list of figures that have associated raw data
- A description of any restrictions on data availability

> GWAS summary statistics will be shared upon publication of the article at http://egg-consortium.org/placental-weight-2023.html.

# Field-specific reporting

Please select the one below that is the best fit for your research. If you are not sure, read the appropriate sections before making your selection.

☒ Life sciences          ☐ Behavioural & social sciences          ☐ Ecological, evolutionary & environmental sciences

For a reference copy of the document with all sections, see nature.com/documents/nr-reporting-summary-flat.pdf

# Life sciences study design

All studies must disclose on these points even when the disclosure is negative.

| | |
|---|---|
| Sample size | In a GWAS meta-analysis, the more samples included, the more GWAS-significant loci will be detected. |
| Data exclusions | Gestational week lower than 37 weeks and over 42 weeks |
| Replication | This study is a meta-analysis. The results have not been replicated in external studies. |
| Randomization | This study is observational, so randomisation is not applicable to this type of study. |
| Blinding | This study is observational, so blinding is not applicable to this type of study. |

# Reporting for specific materials, systems and methods

We require information from authors about some types of materials, experimental systems and methods used in many studies. Here, indicate whether each material, system or method listed is relevant to your study. If you are not sure if a list item applies to your research, read the appropriate section before selecting a response.

### Materials & experimental systems

| n/a | Involved in the study |
|---|---|
| ☒ ☐ | Antibodies |
| ☒ ☐ | Eukaryotic cell lines |
| ☒ ☐ | Palaeontology and archaeology |
| ☒ ☐ | Animals and other organisms |
| ☒ ☐ | Human research participants |
| ☒ ☐ | Clinical data |
| ☒ ☐ | Dual use research of concern |

### Methods

| n/a | Involved in the study |
|---|---|
| ☒ ☐ | ChIP-seq |
| ☒ ☐ | Flow cytometry |
| ☒ ☐ | MRI-based neuroimaging |

