## [Peer Review File · Nature Genetics]

Peer Review Information

Manuscript Title: Genome-wide association study of placental weight identifies distinct and shared genetic influences between placental and fetal growth

Corresponding author name(s): Professor Pal Rasmus Njolstad

Reviewer Comments & Decisions:

Decision Letter, initial version:
--

21st Dec 2022

Dear Professor Njolstad,

Your Article entitled "Genome-wide association study of placental weight in 179,025 children and parents reveals distinct and shared genetic influences between placental and fetal growth" has now been seen by 3 referees, whose comments are attached. While they find your work of potential interest, they have raised serious concerns which in our view are sufficiently important that they preclude publication of the work in Nature Genetics, at least in its present form.

While the referees find your work of some interest, they raise concerns about the strength of the novel conclusions that can be drawn at this stage.

In brief, Reviewers #2 and #3 are broadly positive and supportive of publication; they do not raise any concerns that would require a major revision. Referee #3 has some minor comments.

However, Reviewer #1 submits a report highlighting a fundamental issue with your study: they argue that the PW phenotyping that you conduct the study on has major, multiple issues. The review goes into substantial detail on each of their specific concerns regarding e.g. standardisation of phenotyping as well as the analysed statistics, suggesting that the phenotypes analysed here are not consistent with other studies. That said, they also provide some suggestions for improvement and sound, overall, willing to consider a suitable revision.

Given this concern of PW measurement is fundamental to your entire study, we think that Referee #1's report justifies a rejection. However, given the support of the other referees, we would be interested in a submission that comprehensively addresses this phenotyping issue.

Should further data allow you to fully address these criticisms we would be willing to consider an appeal of our decision (unless, of course, something similar has by then been accepted at Nature Genetics or appeared elsewhere). This includes submission or publication of a portion of this work someplace else.

The required new experiments and data include, but are not limited to those detailed here. We hope you understand that until we have read the revised manuscript in its entirety we cannot promise that it will be sent back for peer review.

If you are interested in attempting to revise this manuscript for submission to Nature Genetics in the future, please contact me to discuss a potential appeal. Otherwise, we hope that you find our referees' comments helpful when preparing your manuscript for resubmission elsewhere.

Sincerely,

Michael Fletcher, PhD
Senior Editor, Nature Genetics
ORCID: 0000-0003-1589-7087

Referee expertise:

Referee #1: reproductive medicine.

Referee #2: precision medicine; computational biology/genomics; fertility.

Referee #3: paediatrics; epidemiology; genetics.

Reviewers' Comments:

Reviewer #1:

Remarks to the Author:

The authors have principally performed a meta-analysis of 21 European GWA studies to assess the maternal, paternal and fetal genetic contribution to placental growth, using placental weight (PW) at term delivery as a proxy. They have also conducted a sub-analysis of mother-father-child trios in one of the studies (MoBa) to investigate the relationship between maternal and fetal effects on PW.

In reviewing this paper, given my expertise and experience, I have focused exclusively on the issue of phenotyping. In my opinion, there are sufficient concerns regarding how the PW data were collected and their accuracy as to lead me to question the validity of conducting a GWAS for this particular measure.

1. The authors have separately published reference charts for PW for gestational age, sex and parity using MoBa data.[1] In that paper (which is not referenced in the present paper) male PWs were consistently greater than female PWs but the influence of parity on PW was far greater than that exerted by sex, and the difference was dependent on gestational age. For example, for males, the mean PW differences at the 50th and 90th centiles between multiparous and nulliparous mothers were 35.9g and 46.6g, respectively.[1]

In the present paper, however, parity is only mentioned in some Supplemental Tables and Figures (all unlabelled), which seem identical to the MoBa tables and figures published in the reference chart paper.[1] Lastly, in 9/21 (43%) of the studies the female PW was greater than the male PW - in one study (GOYA-CH-OW) by as much as 25g.

The authors would need to explain why these sex discrepancies are at odds with the published literature and their own reference charts; why they have not incorporated parity as a relevant factor, and why they seemingly generated z-scores from the present PW data (line 607) rather than use the

z-scores generated from their reference charts. They would also need to explain the Supplemental Tables and Figures provided.

2. I have other concerns about the PW data:

a) The 21 studies contributing to the meta-analysis collected data from 1952 to 2013, and five (24%) of the studies finished collecting data more than 30 years ago. Over the course of those 70 years, the way in which gestational age is estimated has changed markedly: from reliance on the date of the (often uncertain) last menstrual period (LMP) to ultrasound measurement of the fetus in the first trimester as the gold standard.

Thus, for example, the ALSPAC Study only used the estimated date of delivery based on the LMP,[2] whilst in the Generation R Study, pregnancy dating was performed using the first ultrasound measurement of crown-rump length (CRL) or biparietal diameter (BPD), with dating curves derived from this cohort.[3] The lack of consistency across studies in estimating gestational age at birth is not a criticism of the studies themselves. I simply think the authors should acknowledge the large heterogeneity in the dataset, which affects their calculated z-scores.

b) What I believe is more important is the lack of standardisation in the way in which PW was recorded from a variety of sources across these studies, which was inevitable given that PW was collected as part of routine clinical care. The factors that influence PW, which appear not to have been considered in the present study, are the way in which the placenta is processed (trimmed v. untrimmed); the timing of cord clamping (early v. late); how soon after delivery the placenta is measured (early v. late), and the mode of delivery (vaginal v. Caesarean). In a well-designed prospective study, with a focus of PW, one would expect these factors to be standardised, as well as the accuracy of the scales (mechanical v. digital) and how often they are calibrated.

The issue is not inconsequential, as recognised in reviews,[4, 5] given the effect of these factors on PW. For example, in a well-conducted, prospective research study, the placentas should be trimmed of the fetal membranes and umbilical cord (after removing blood clots) and weighed soon after delivery. Having said that, the trimmed placental disc weight may be estimated by subtracting 16% (i.e., weight of the fetal membranes and umbilical cord) from the untrimmed placental weight,[6] although in another study the weight difference was 16.1-41.5%.[7]

Leary et al. also found a 16% median difference between untrimmed and trimmed weight, but the mode of delivery had a significant effect on this difference: medians for vaginal and Caesarean section deliveries were 19% and 14%, respectively, because of differences in the length of umbilical cord cut at the time of delivery.[8] One assumes that the placentas were all untrimmed in the present study but there remains the possibility that some were not. Unless the authors know which is the case, this possibility should be acknowledged as well as the inconsistencies likely due to the lack of standardisation in measuring PW.

The timing of weighing the placenta is also important as stored placentas (kept refrigerated but unfixed) lose approximately 5% of their weight in the first 12-24 hours, and 10% over 48 hours, which may reflect loss of blood from tears.[5]

Whilst the differences mentioned above may be small, the timing of cord clamping has a major effect on PW related to the amount of blood passing to the baby. Bouw et al. (1976) reported mean

placental weights of 567.4g (SD 114.5g) and 407.1g (SD 74.0g) associated with early and late cord clamping, respectively.[4] Delayed cord clamping has been practised since at least the late 1950's,[9] but its possible effect on this dataset is unknown and not acknowledged here.

Burton et al. (2004) summarised perfectly the problem of a lack of standardisation: "If we assume an average placental weight of around 500g at term, the variation may be up to 20% depending on timing of cord clamping, and around 15% depending on whether complete or membrane-trimmed, with small effects of maternal characteristics, delivery mode and storage time".[5]

The authors themselves acknowledged many of these issues in their reference chart paper when referring to other studies: "All three studies weighed untrimmed placentas, however variations between the studies could also be partially due to the timing, differences in collection methods of the placenta (e.g. removal of clots etc.) and in the case of Wallace et al. rounding to the nearest 10g".[1]

3. Many of my concerns about the lack of standardisation in measuring PW apply equally to the measurement of birth weight. In addition, birth weight has been treated in these analyses as an absolute value, which should have been more meaningfully transformed into a z-score against international standards for gestational age and sex.

4. In the discussion regarding the association between PW and preeclampsia, I agree that the findings are 'counter-intuitive' (line 565). The argument presented would undoubtedly be improved by reporting any differences between the early and late onset cases in this dataset. Having said that, the sample size in the analysis of the MoBa trios is not great and there is an argument for assigning the analysis of these phenotypes, including preeclampsia, to a separate paper.

I also question the choice of blood pressure and neuropsychiatric traits. The latter because the literature is so controversial and mixed, the former because of the choice of systolic and diastolic blood pressure as phenotypes. Which values during pregnancy have been used? Those at the first antenatal clinic visit? The maximum recorded at any point in pregnancy? On anti-hypertensive treatment or not? The point is that the use of routinely collected blood pressure measurements, without any degree of standardisation within and across studies, is notoriously unreliable.

Minor issues

1. The title itself is a slight misnomer and does not accurately portray the fact that only 65,405 newborns (not children) were genotyped.
2. 'Miscarriage' should replace 'spontaneous abortion' (line 165).
3. The trios are described as 'child-mother-father' and 'mother-father-child'; consistency would be preferable.
4. 21 studies contributed to the meta-analysis, not 28 (line 178).
5. Although the information is provided in the Supplementary Tables, it is unclear in the paper itself which of the 21 studies contributed to the maternal and paternal GWAS.
6. In various places within the manuscript, the acronym PW is not used.
7. The North Finland Birth Cohort recorded PW at birth, not from a maternal questionnaire as indicated in Supplementary Table 1.
8. The PDF of Supplementary Table 1 is difficult to read and would benefit from being printed in landscape format.
9. In the separate analysis of mother-father-child trios, what is the difference between 'nausea and vomiting in pregnancy' and 'hyperemesis'?

References

1. Flatley, C., et al., Placental weight centiles adjusted for age, parity and fetal sex. *Placenta*, 2022. 117: p. 87-94.
2. Golding, J., et al., ALSPAC--the Avon Longitudinal Study of Parents and Children. I. Study methodology. *Paediatr Perinat Epidemiol*, 2001. 15(1): p. 74-87.
3. Gaillard, R., et al., Individually customised fetal weight charts derived from ultrasound measurements: the Generation R Study. *Eur J Epidemiol*, 2011. 26(12): p. 919-26.
4. Bouw, G.M., et al., Quantitative morphology of the placenta 1. Standardization of sampling. *Eur J Obstet Gynecol Reprod Biol*, 1976. 6(6): p. 325-331.
5. Burton, G.J., et al., Optimising sample collection for placental research. *Placenta*, 2014. 35(1): p. 9-22.
6. Ma, L.X., D. Levitan, and R.N. Baergen, Weights of Fetal Membranes and Umbilical Cords: Correlation With Placental Pathology. *Pediatr Dev Pathol*, 2020. 23(4): p. 249-252.
7. Tan, K.L., et al. Weighing practices and their effect on the placental-baby weight ratio. in 16th World Congress in Fetal Medicine. 2017. Ljubljana, Slovenia.
8. Leary, S.D., et al., Contribution of the umbilical cord and membranes to untrimmed placental weight. *Placenta*, 2003. 24(2-3): p. 276-8.
9. Nyberg, R. and B. Westin, [Residual blood volume of placenta after delayed clamping of cord]. *Nord Med*, 1958. 59(12): p. 441-2.

Reviewer #2:

Remarks to the Author:

Using placental weight after term delivery as a proxy for placental growth, the authors report genome-wide association analyses in fetal, maternal, and paternal genomes, yielding 40 independent association signals, which some being classified as fetal only and only a few being classified as maternal or paternal only. A number of comprehensive analyses are carried out to further interrogate the genetic associations including genetic correlation and colocalization analyses revealing overlap with birth weight genetics. Several loci were classified as only affecting placental weight, with connections to placental development and morphology, and transport of antibodies and amino acids. Mendelian randomization analyses indicate that higher placental weight is causally associated with risk of preeclampsia or shorter gestational duration and identify a relationship between insulin produced by the fetus and the growth of the placenta, providing a key link between fetal and placental growth. The study is comprehensive, the analyses are very robust and manuscript is well written. While the clinical significance of placental growth itself is not as clear at first and might detract from the general interest, links between placental growth and some of the adverse maternal and fetal outcomes that are explored later in the manuscript are very interesting.

Reviewer #3:

Remarks to the Author:

This paper reports the first 'successful' GWAS for placental weight (PW), finding 40 independent loci. Although it includes no replication sample, the findings have coherent effects on birth weight (BW), signals are classified as (mostly) fetal vs parentally acting, and lead to interesting functional and causal insights.

Comments

Line 243 "a small component from the paternal genome ($h^2=0.06$ ($SE=0.02$))" - clarify if this estimate is conditioned on (i.e. independent of) fetal genome? If so, is this surprising? What might be the mechanism for an independent paternal genotype effect on PW? Also clarify if this analysis (in lines 239 to 243) is restricted to the trios?

Lines 318 to 328 Correlations between PW and BW - this section is unnecessarily long and confuses BW vs PW comparisons with fetal vs parental comparisons which have already been described. Restrict to PW vs BW correlations for each genome source.

Figures 3a and 5 are confusing and do not support the results, likely because they both unhelpfully combine fetal vs parental AND 'PW-only' vs 'PW & BW' comparisons:

Fig 3a - Is meant to demonstrate the discrimination between fetal vs parental effects but does not achieve this. For some 'fetal' genes (e.g. RPL31P11, ARHGAP26, LOC339593) the MT (fetal) effect does not seem larger than the MnT (maternal) effect. CHRM3 (labelled 'ambiguous') appears to have a much larger MT than MnT effect.

Furthermore, the classification of loci as PW-only or 'PW & BW' appears to be inconsistent with the displayed effects on PW and BW - possibly because this MoBa-only sample was not used for this classification (Fig 3 is not cited in that description on page 13).

Fig 5 is also too complex and fails to demonstrate the discrimination between the 12 PW-only signals and the 28 'PW & BW' signals. Addition of fetal vs parental comparisons confuses this aim.

Fig 4 - Clarify that the Y axis label indicates "Rg with PW"

Fig 6 - It seems that the Y-axis labels for maternal and fetal genomes are erroneously swapped

Line 380 - the 31 fetal PW genes ranked higher (than other genes) in maternal innate lymphocyte cells ($P = 1.2 \times 10^{-3}$). What is the explanation for this - i.e. how can fetal genotype alter gene expression in maternal cells?

Interesting functional insights are provided from analyses of mRNA expression and methylation. It would be helpful to cite a reference for the stated role of SLC7A5 in the placenta (Line 415)

Line 508 - PW and later neuropsychiatric traits - cite evidence for the link between these traits, and state which direction has been reported? Is the observed relationship with ADHD consistent with literature? Are genetic correlations seen? Line 512-515, confirmation of the PW-PRS association with PW in these patient cohorts is expected and could be moved to the Methods.

Line 542-559 - Discussion on KCNQ1 ignores the finding here that this signal is classified as 'PW-only'. Instead this section rationalises for various effects of KCNQ1 on both fetal and placental growth. This should be corrected / explained.

Line 573-577 - FLT1 is not related to PW so has low relevance here. This section should try to rationalise why higher PW is associated with preeclampsia risk.

Line 747 - Enrichment analysis Methods "Significance levels for this enrichment analysis were Bonferroni corrected...". State the corrected thresholds.

Decision Letter, Appeal:

5th Apr 2023

Dear Pål,

Thank you for your message asking us to reconsider our decision on your manuscript "Genome-wide association study of placental weight in 179,025 children and parents reveals distinct and shared genetic influences between placental and fetal growth".

I have now discussed the points of your letter with my colleagues, and we think that your revision has addressed the major concerns that we highlighted from the initial round of review, most notably the questions on phenotyping. We therefore invite you to resubmit your manuscript for review by the original referees.

When preparing a revision, please ensure that it fully complies with our editorial requirements for format and style; details can be found in the Guide to Authors on our website (<http://www.nature.com/ng/>).

Please be sure that your manuscript is accompanied by a separate letter detailing the changes you have made and your response to the points raised. At this stage we will need you to upload:

1) a copy of the manuscript in MS Word .docx format.

2) The Editorial Policy Checklist:

<https://www.nature.com/documents/nr-editorial-policy-checklist.pdf>

3) The Reporting Summary:

(Here you can read about the role of the Reporting Summary in reproducible science:

<https://www.nature.com/news/announcement-towards-greater-reproducibility-for-life-sciences-research-in-nature-1.22062>)

Please use the link below to be taken directly to the site and view and revise your manuscript:

[redacted]

With kind wishes,

Michael Fletcher, PhD

Senior Editor, Nature Genetics

ORCID: 0000-0003-1589-7087

Author Rebuttal to Initial comments

related to the phenotyping issue. Please see specific responses below.

Reviewer #1

The authors have principally performed a meta-analysis of 21 European GWA studies to assess the maternal, paternal and fetal genetic contribution to placental growth, using placental weight (PW) at term delivery as a proxy. They have also conducted a sub-analysis of mother-father-child trios in one of the studies (MoBa) to investigate the relationship between maternal and fetal effects on PW.

In reviewing this paper, given my expertise and experience, I have focused exclusively on the issue of phenotyping. In my opinion, there are sufficient concerns regarding how the PW data were collected and their accuracy as to lead me to question the validity of conducting a GWAS for this particular measure.

RESPONSE

We thank the reviewer for this thorough assessment of our work. As detailed below, we have carried out further analyses and extended the text in response to points raised by the reviewer.

1. The authors have separately published reference charts for PW for gestational age, sex and parity using MoBa data.[1] In that paper (which is not referenced in the present paper) male PWs were consistently greater than female PWs but the influence of parity on PW was far greater than that exerted by sex, and the difference was dependent on gestational age. For example, for males, the mean PW differences at the 50th and 90th centiles between multiparous and nulliparous mothers were 35.9g and 46.6g, respectively.[1]

In the present paper, however, parity is only mentioned in some Supplemental Tables and Figures (all unlabelled), which seem identical to the MoBa tables and figures published in the reference chart paper.[1] Lastly, in 9/21 (43%) of the studies the female PW was greater than the male PW - in one study (GOYA-CH-OW) by as much as 25g.

The authors would need to explain why these sex discrepancies are at odds with the published literature and their own reference charts; why they have not incorporated parity as a relevant factor, and why they seemingly generated z-scores from the present PW data (line 607) rather than use the z-scores generated from their reference charts. They would also need to explain the Supplemental Tables and Figures provided.

RESPONSE

We thank the reviewer for the detailed review of the cohort characteristics and for raising these four concerns:

1 Sex differences

We agree with the reviewer that the point estimate of the mean placental weight (PW) is higher for females than males in some of the relatively smaller cohorts (between 300 and 1,000 pregnancies). However, the standard deviation of PW is large (consistently between

120-160 g in most studies). This means that for the smaller studies, the confidence intervals (CIs) around the point estimate of the mean PW are wide and overlap between sexes. We have now included formal tests for the difference in PW between females and males (added to Supplementary Tables 1, 3 and 5). After accounting for multiple testing, none of the *t*-tests in the smaller cohorts showed evidence of a difference between male and female PW, whereas in the larger cohorts, where there was greater power to detect small differences in mean PW (iPSYCH, MoBa, DBDS, Hunt, NFBC1986, and ALSPAC), the *t*-tests with $P < 0.05$ consistently indicated higher PW in males compared with females. In response to the reviewer's comment, we went back to the contributing cohorts and found that for the GOYA and Roskilde cohorts the means for males and females had inadvertently been swapped around, which has now been corrected. All other cohorts confirmed that the values in the summary tables were correct. Please note that this correction only concerns the tables of descriptives and does not affect our results.

2 Parity

We have indeed, in a previous study based on the MoBa cohort, found a significant association between parity and PW (Flatley et al., 2022, PMID: 34773745). However, in constructing the analysis plan for the current study, we made the decision (as we have done previously in genome-wide association [GWA] studies of birth weight [BW]) to make minimal adjustments for covariates in the analysis, selecting only sex, gestational age (GA) and ancestry principal components, as these were readily available in all studies. Adjustment for population substructure within each cohort using informative principal components helps to guard against confounding in GWAS: spurious associations may arise when alleles of a genetic variant differ in frequency between population subgroups who also have different mean PWs.

As the reviewer points out, conditioning on another phenotype like parity would reduce phenotypic variation, hence increasing power to detect genetic association, but it would also remove potentially interesting association signals with PW that are mediated by parity. In general, adjusting for heritable traits is known to be a source of bias in GWA studies, and it is recommended to keep such covariates to a minimum (Aschard et al., 2015, PMID: 25640676). Instead, it is recommended to conduct downstream analyses to investigate whether a given association signal is mediated by another phenotype or presents signs of pleiotropy. That is what we did for GA. We hope that better powered and more detailed analyses will be able to build on our work to investigate the relationship of our results with parity.

In response to the reviewer's comment, we have conducted a number of sensitivity analyses in the MoBa cohort to assess the robustness of our results. Specifically, we performed association analyses for the top SNP at all GWA significant loci in MoBa children, mothers, and fathers with statistical models that in addition to ten principal components (PCs) included the following sets of covariates: (1) none, (2) sex, (3) sex + GA, (4) sex + GA + parity. These results are included in Figure A below and attached Table A, and as can be seen, the inclusion of parity had minimal influence on the estimated effects and CIs for the top SNPs.

Figure A: Effect size estimates and 95% confidence intervals for the association with PW of the different signals in 19,861 unrelated trios in MoBa using different sets of covariates in addition to the batches and principal components: (1) none, (2) sex, (3) sex + GA, (4) sex + GA + parity

We agree with the reviewer that potential genetic influences on differences in placentation between pregnancies of different parity would be interesting to investigate, but we feel it would be beyond the scope of the current study. We would be happy to include the above as supplementary material if the Editor feels it would be necessary.

3 Z-scores, reference charts, and inter-cohort heterogeneity

The reason why we chose not to use the reference centiles computed in MoBa to calculate the z-scores, was mainly due to the heterogeneity in study designs (population based vs. ascertained), as well as potential differences in phenotype definition among cohorts (see responses to the reviewer's comments for a fuller discussion on this point). Asking all cohorts to standardize their values according to MoBa, which is part of the study, could have introduced a bias towards MoBa in the meta-analysis. By asking for Z-scores to be calculated based on the mean PW for each study, we aimed to circumvent between-study differences in measurement: under the fair assumption that the respective policy at the time and place of collection was adhered to, whether the placenta was trimmed in some studies and untrimmed in others should not present a problem because the Z-score is a measure of each individual PW in relation to the mean weight for that study.

In response to the concerns raised about the influence of the putative heterogeneity in measurements and standardization of PW, we have now included a more comprehensive set of figures and analyses that we believe will allow the readers to better assess the consistency *between* studies.

First, we have provided the Forest plots from the meta-analysis for the identified lead SNPs from all 41 association signals (Supplementary Figures 15a-h) to give a visual illustration of the SNP effect estimates across studies. These plots indeed illustrate that the effect estimates are consistent across studies, and importantly that no single study consistently diverges from the general pattern - indicating that the phenotype measurements and study design allow for the identification of robust PW-SNP associations.

Second, we provide the formal heterogeneity test as part of our meta-analysis, both in the new figures and in Supplementary Table 7. Only two of the 41 lead variants showed some evidence of heterogeneity ($P = 0.03$ and 0.04). We cannot rule out that there is some heterogeneity in SNP effects between cohorts for these two SNPs. However, it should be noted that this number is also consistent with chance (2 in 41). Moreover, these two variants are relatively rare, with minor allele frequencies of 6.6% and 0.7%, respectively; making them more vulnerable to stochastic variation.

Overall, these analyses suggest that the genetic associations identified are robust, with little between-study heterogeneity in genetic associations, despite the between-study heterogeneity in phenotyping. The statistics for these tests and a graphical representation of forest plots of the estimates for each study and the meta-analysis are now reported as part of the supplementary material (Supplementary Table 7, Supplementary Figures 15a-h), and we have added a line in the Results (paragraph 1, page 6) and a paragraph to the Discussion (paragraph 3, page 24) to address the concerns.

4 Explain supplementary figures and tables

We do not fully understand the concern of the reviewer on this point: all our supplementary tables and figures were labeled with a detailed legend, in which we described the content (e.g. of every table column) . Some of the supplementary figures are very large and the figure caption might be on a different page from the figure. The supplementary tables were labelled within each sheet of the excel file. This information may have been lost through conversion in the manuscript tracking system, but the source files should be available for download through the system (see answers to minor comment #8). We checked again all tables and figures and would be happy to implement any specific concern of the reviewer or Editor. If, despite our efforts, some files do not comply with the policy of the journal, we will of course ensure that all figures and tables are in a format that is compatible with *Nature Genetics* standards.

2. I have other concerns about the PW data:

a) The 21 studies contributing to the meta-analysis collected data from 1952 to 2013, and five (24%) of the studies finished collecting data more than 30 years ago. Over the course of those 70 years, the way in which gestational age is estimated has changed markedly: from reliance on the date of the (often uncertain) last menstrual period (LMP) to ultrasound measurement of the fetus in the first trimester as the gold standard.

Thus, for example, the ALSPAC Study only used the estimated date of delivery based on the LMP,[2] whilst in the Generation R Study, pregnancy dating was performed using the first ultrasound measurement of crown-rump length (CRL) or biparietal diameter (BPD), with dating curves derived from this cohort.[3] The lack of consistency across studies in estimating gestational age at birth is not a criticism of the studies themselves. I simply think the authors should acknowledge the large heterogeneity in the dataset, which affects their calculated z-scores.

RESPONSE

The reviewer is right that changes in GA estimation methods from the use of LMP to ultrasound measurement are likely to result in some heterogeneity among cohorts included in the meta-analysis. We do, however, believe that within our cohorts, which are limited to term births (37- 42 weeks of gestation), this heterogeneity will have limited effect on our association results. In a paper published by several authors of the current study, we used data from the Swedish National Birth Register and found a difference in the median GA of only five days for measures taken between 1973 - 2012 (3,940,577 pregnancies) (Modzelewska et al., 2020, PMID: 33156833). This median is taking into account all births between 154 - 301 days. As our current study was performed only on term pregnancies, this difference would be less. Furthermore, given that the vast majority (~85%) of samples in our recent cohort were collected during the use of ultrasound based dating, any potential effects of the GA estimation heterogeneity on the SNP-PW association results would be negligible. In addition to these points, the high concordance between the results from our sex-only adjusted meta-analysis and the sex- and GA-adjusted analysis shows that the SNP-PW associations are very similar whether or not GA is included as a covariate (see genetic correlation results [higher than 0.99] in the Results section and the Fig. A shown above), meaning that any potential impact of heterogeneity in accuracy of GA measurement on our

analysis is small. We do understand the importance of clarifying this in the Discussion, which we have extended accordingly (paragraph 3, page 24). In regard to the effect on Z-scores, the PW Z-scores did not adjust for GA, rather this was performed within the regression, as clearly stated in the Methods (paragraph 1, page 25).

b) What I believe is more important is the lack of standardisation in the way in which PW was recorded from a variety of sources across these studies, which was inevitable given that PW was collected as part of routine clinical care. The factors that influence PW, which appear not to have been considered in the present study, are the way in which the placenta is processed (trimmed v. untrimmed); the timing of cord clamping (early v. late); how soon after delivery the placenta is measured (early v. late), and the mode of delivery (vaginal v. Caesarean). In a well-designed prospective study, with a focus of PW, one would expect these factors to be standardised, as well as the accuracy of the scales (mechanical v. digital) and how often they are calibrated.

The issue is not inconsequential, as recognised in reviews,[4, 5] given the effect of these factors on PW. For example, in a well-conducted, prospective research study, the placentas should be trimmed of the fetal membranes and umbilical cord (after removing blood clots) and weighed soon after delivery. Having said that, the trimmed placental disc weight may be estimated by subtracting 16% (i.e., weight of the fetal membranes and umbilical cord) from the untrimmed placental weight,[6] although in another study the weight difference was 16.1-41.5%.[7]

Leary et al. also found a 16% median difference between untrimmed and trimmed weight, but the mode of delivery had a significant effect on this difference: medians for vaginal and Caesarean section deliveries were 19% and 14%, respectively, because of differences in the length of umbilical cord cut at the time of delivery.[8] One assumes that the placentas were all untrimmed in the present study but there remains the possibility that some were not. Unless the authors know which is the case, this possibility should be acknowledged as well as the inconsistencies likely due to the lack of standardisation in measuring PW.

The timing of weighing the placenta is also important as stored placentas (kept refrigerated but unfixed) lose approximately 5% of their weight in the first 12-24 hours, and 10% over 48 hours, which may reflect loss of blood from tears.[5]

Whilst the differences mentioned above may be small, the timing of cord clamping has a major effect on PW related to the amount of blood passing to the baby. Bouw et al. (1976) reported mean placental weights of 567.4g (SD 114.5g) and 407.1g (SD 74.0g) associated with early and late cord clamping, respectively.[4] Delayed cord clamping has been practised since at least the late 1950's,[9] but its possible effect on this dataset is unknown and not acknowledged here.

Burton et al. (2004) summarised perfectly the problem of a lack of standardisation: "If we assume an average placental weight of around 500g at term, the variation may be up to 20% depending on timing of cord clamping, and around 15% depending on whether complete or membrane-trimmed, with small effects of maternal characteristics, delivery

mode and storage time”.[5]

The authors themselves acknowledged many of these issues in their reference chart paper when referring to other studies: “All three studies weighed untrimmed placentas, however variations between the studies could also be partially due to the timing, differences in collection methods of the placenta (e.g. removal of clots etc.) and in the case of Wallace et al. rounding to the nearest 10g”.[1]

RESPONSE

We agree with the reviewer that accounting for such differences is important in a well-designed prospective study. However, the design of a GWA meta-analysis like ours must balance two conflicting priorities: minimize phenotype-based heterogeneity and maximize sample size (Evangelou and Ioannidis, 2013, PMID: 23657481). The history of GWA studies has demonstrated that sample size is the primary driver of discovery (Abdellaoui et al., 2023, PMID: 36634672). We therefore attempted to include as many cohorts as possible and control for heterogeneity where possible, in accordance with the standards in the field. Further, we conducted the GWA meta-analysis according to the reference statistical methods and tools of the field proven to be robust towards heterogeneity, and performed strict quality control at both cohort and SNP level, following best practices in GWA meta-analyses (Begum et al., 2012, PMID: 22241776).

We agree with the reviewer that our study would have increased sensitivity with more accurate and harmonized phenotypes across cohorts, but such data are not available in a sample with sufficient power to detect the characteristically small effects of common genetic variants. Notably, although phenotypic heterogeneity will reduce the statistical power to detect associations (compared with a scenario with the same sample size where the trait was recorded in a standardized uniform way in all cohorts), it will not lead to spurious association signals because alleles are distributed in the population independently of placenta collection and weighing methods. Hence, the potential drawback with a more heterogeneous phenotype is outweighed by the huge gain in power made possible by increasing the sample size. Furthermore, heterogeneity is assessed and accounted for in meta-analyses. It is in fact a strength of our meta-analyses that findings are robust towards data collection practices across time and measurement sites. We are therefore confident that the key findings of our study (the 40 identified loci) have robust and valid supporting evidence.

There are some examples we would like to mention:

- 1) Previous large GWA meta-analyses have included data from the UK Biobank, where the BW of the first child of women in the study was self-reported to the nearest pound. When the UK Biobank GWAS results for identified loci were compared with those from cohorts with detailed phenotypes using birth registry data, the results were highly concordant (Juliusdottir et al., 2021, PMID: 34282336; Warrington et al., 2019, PMID: 31043758). In fact, the larger sample sizes in the UK Biobank helped in discovering more *bona fide* variants than the smaller more uniformly phenotyped samples.

2) In the first major GWAS of GA (Zhang et al., 2017, PMID: 28877031), GA was self-reported in the discovery cohort. This was complemented in the replication cohorts using accurate GA data from Nordic birth registers. The heterogeneous discovery cohort data was replicated in the homogenous register data.

To further address the reviewer's specific concern regarding the method of measuring PW, we would like to provide more details on the method of measuring PW in MoBa, the largest cohort of our meta-analysis, which we have used for several of the follow-up analyses. Data related to pregnancy and birth in MoBa are standardized and stem from the Medical Birth Registry of Norway (MBRN). Reporting PW to the MBRN is mandatory, and is carried out by the midwife attending the birth. In Norway, all midwives share curriculum and training regarding the reporting of data, including examination of the afterbirth to the MBRN. The placenta is examined and characteristics of the placenta and cord, including measurements of the PW (untrimmed with the cord and membranes attached) is reported. The method has been unchanged since the inception of the MBRN in 1967, although the report form was altered from free text to tick boxes regarding categorical variables in 1999. The reporting of these data to the MBRN has been validated, with good inter- and intra-observer agreement, making the data suitable for large scale epidemiological research (Sunde et al., 2017, PMID: 28481411). This validation study was carried out in 2013 and 2016, in different institutions, involving many different midwives, and both high- and low-risk wards, ensuring representativity for different settings involved in MoBa. The inter-observer agreement in measurement of PW was good; the mean PW differed by -3.8 g (measured by the blinded colleague, compared with the attending midwife). The findings indicate that the data from the institutions transferred to the MBRN on PW have a good precision and reliability.

3. Many of my concerns about the lack of standardisation in measuring PW apply equally to the measurement of birth weight. In addition, birth weight has been treated in these analyses as an absolute value, which should have been more meaningfully transformed into a z-score against international standards for gestational age and sex.

RESPONSE

For the sake of comparability with the results on PW, the BW analyses on MoBa presented in Figure 3, Supplementary Figure 5, and in Supplementary Table 11 were transformed into a Z-score (using the within-cohort mean and SD of BW, i.e. each individual's Z-score was generated as $[BW - \text{meanBW}] / \text{SDBW}$, in exactly the same way as for PW). We apologize that this was unclear from the text, the Methods section (paragraph 3, page 27) has been updated to clarify this point.

All other results on BW were taken from the most recent GWA meta-analysis by Juliusdottir et al., 2021 (PMID: 34282336). In that publication, and in all other previous GWA meta-analyses of BW (eg. Warrington et al., 2019, PMID: 31043758), the BW phenotype was standardized within each study, and then adjusted for sex and GA, by including those variables as covariates in the regression analysis, where available.

4. In the discussion regarding the association between PW and preeclampsia, I agree that the findings are 'counter-intuitive' (line 565). The argument presented would undoubtedly be improved by reporting any differences between the early and late onset cases in this dataset. Having said that, the sample size in the analysis of the MoBa trios is not great and there is an argument for assigning the analysis of these phenotypes, including preeclampsia, to a separate paper.

I also question the choice of blood pressure and neuropsychiatric traits. The latter because the literature is so controversial and mixed, the former because of the choice of systolic and diastolic blood pressure as phenotypes. Which values during pregnancy have been used? Those at the first antenatal clinic visit? The maximum recorded at any point in pregnancy? On anti-hypertensive treatment or not? The point is that the use of routinely collected blood pressure measurements, without any degree of standardisation within and across studies, is notoriously unreliable.

RESPONSE

We would like to clarify that we do not believe the findings regarding the association between PW and preeclampsia are counter-intuitive. Our intention was to indicate in the text that the findings may initially seem counter-intuitive, given that preeclampsia can be associated with growth restriction. However, we believe our finding is consistent with the model of term preeclampsia which includes crowding of the intervillous space and subsequent uteroplacental mismatch, which is outlined in the recent review by Magee et al., 2022 (PMID: 35544388). We have revised the Discussion (paragraph 2, pages 23) to try and make this clearer. We would also like to clarify that we did not use the MoBa trios for the preeclampsia analysis. The Mendelian randomization analysis was performed using our GWA meta-analysis results for PW (exposure) and GWA data from the largest available fetal and maternal meta-analysis of preeclampsia (outcome). We were thus limited to the definition of preeclampsia in that study (Steinthorsdottir et al. 2020, PMID: 33239696), which did not separate early and late cases. In our study, most cases of early-onset preeclampsia would be excluded by the exclusion criteria. We agree with the reviewer that it would be preferable to perform dedicated analyses on early- and late-onset preeclampsia. As the reviewer notes, however, power is an issue and a stratified analysis is beyond the scope of our study. We still feel that the result is an important finding, and that it should be included in the current paper.

The rationale behind the choice of the analysis of blood pressure is its known relationship with BW (see for example Juliusdottir et al., 2021, PMID: 34282336). As above, the data used to perform these analyses come from large GWA meta-analyses of blood pressure, meaning that the SNP-blood pressure associations were identified in non-pregnant women and men. However, we have previously shown that they do capture blood pressure in pregnancy (Tyrrell et al., 2016, PMID: 26978208). We agree with the reviewer regarding the difficulty to assess blood pressure during pregnancy, but we did not perform observational analyses of blood pressure measurement associations with PW. Rather, we used genetic instruments to proxy for blood pressure measurements in a Mendelian randomization analysis (see Methods, paragraph 1, page 30). Genetic variants are unlikely to be subject to confounding or reverse causation than direct measurements, and they reflect long-term

values, rather than measurements at a particular time (see Davies et al., 2018, PMID: 30002074).

The rationale behind the choice of investigating neuropsychiatric traits is indeed because literature on the topic is so controversial and mixed. The text has been extended to clarify this, please see also the answer to Reviewer #3 on this point. We hope that our work will help shed light on this important topic.

Minor issues

1. The title itself is a slight misnomer and does not accurately portray the fact that only 65,405 newborns (not children) were genotyped.

RESPONSE

We have now clarified the title to mention the number of children and parents separately. We agree with the reviewer that using children is imprecise. We have changed to newborns in the title.

2. 'Miscarriage' should replace 'spontaneous abortion' (line 165).

RESPONSE

We have implemented the suggested change.

3. The trios are described as 'child-mother-father' and 'mother-father-child'; consistency would be preferable.

RESPONSE

We thank the reviewer for this suggestion and have changed the text accordingly (using child-mother-father).

4. 21 studies contributed to the meta-analysis, not 28 (line 178).

RESPONSE

We apologise for the confusion. The total number of studies contributing to the fetal meta-analysis was 21, however the number of unique studies contributing to all analyses was 28: 16 studies contributed to the maternal meta-analysis, and six contributed to the paternal meta-analysis, with 11 of these studies contributing to more than one of the meta-analyses. We have now edited the text to clarify (paragraph 4, page 5) and refer to Figure 1, which contains the detailed information on study design.

5. Although the information is provided in the Supplementary Tables, it is unclear in the paper itself which of the 21 studies contributed to the maternal and paternal GWAS.

RESPONSE

We have now extended the text to clarify that some cohorts contributed with parents, children, or both, and point to the relevant tables (paragraph 2, page 6). We feel the information is sufficient, but would be happy to consider moving more detail of the cohorts in each meta-analysis to the main paper if the Editor requests it.

6. In various places within the manuscript, the acronym PW is not used.

RESPONSE

We have now implemented PW throughout the manuscript.

7. The North Finland Birth Cohort recorded PW at birth, not from a maternal questionnaire as indicated in Supplementary Table 1.

RESPONSE

We have changed this fact in Supplementary Table 1.

8. The PDF of Supplementary Table 1 is difficult to read and would benefit from being printed in landscape format.

RESPONSE

We uploaded the supplementary tables as a single Excel file as usually required by the *Nature* journals. We assume that the pdf file is produced by the manuscript tracking system. In the page where files are available for download, the reviewer should be able to download both the Excel file and the source. Two links should be displayed in the following format: Supplementary Table 1 Supplementary Dataset (xxxKB) Source File (XLSX) yyyKB
Alternatively, the supplementary tables are available from our preprint (doi.org/10.1101/2022.11.25.22282723).

9. In the separate analysis of mother-father-child trios, what is the difference between 'nausea and vomiting in pregnancy' and 'hyperemesis'?

RESPONSE

We did not carry out an analysis of 'nausea' and 'vomiting in pregnancy' or 'hyperemesis' in the child-mother-father trios. Instead, we looked up the PW SNP associations in published summary GWA data for those phenotypes (results in Supplementary Figures 10 and 11, and Supplementary Table 16). These analyses were described in the Methods, paragraph 1, page 30, with appropriate references. The phenotypes were defined by Fejzo et al., 2018 (PMID: 29563502; ref. 36 in our main manuscript). 'Nausea' and 'vomiting in pregnancy' were defined as participants' answers to six questions described in Supplementary Note 2 of their paper. Briefly, they were asked if they experienced morning sickness in their pregnancies, and if so, about the severity of the morning sickness. 'Hyperemesis' cases were defined as those additionally reporting receiving intravenous therapy for nausea and vomiting in pregnancy.

Reviewer #2

Using placental weight after term delivery as a proxy for placental growth, the authors report genome-wide association analyses in fetal, maternal, and paternal genomes, yielding 40 independent association signals, which some being classified as fetal only and only a few being classified as maternal or paternal only. A number of comprehensive analyses are carried out to further interrogate the genetic associations including genetic correlation and colocalization analyses revealing overlap with birth weight genetics. Several loci were classified as only affecting placental weight, with connections to placental development and morphology, and transport of antibodies and amino acids. Mendelian randomization analyses indicate that higher placental weight is causally associated with risk of preeclampsia or shorter gestational duration and identify a relationship between insulin produced by the fetus and the growth of the placenta, providing a key link between fetal and placental growth. The study is comprehensive, the analyses are very robust and manuscript is well written. While the clinical significance of placental growth itself is not as clear at first and might detract from the general interest, links between placental growth and some of the adverse maternal and fetal outcomes that are explored later in the manuscript are very interesting.

RESPONSE

We thank the reviewer for the positive response. We are pleased that the data are of interest, and that there are no concerns regarding scientific quality.

Reviewer #3

This paper reports the first 'successful' GWAS for placental weight (PW), finding 40 independent loci. Although it includes no replication sample, the findings have coherent effects on birth weight (BW), signals are classified as (mostly) fetal vs parentally acting, and lead to interesting functional and causal insights.

RESPONSE

We thank the reviewer for this positive comment.

Comments

Line 243 "a small component from the paternal genome ($h^2=0.06$ ($SE=0.02$))" - clarify if this estimate is conditioned on (i.e. independent of) fetal genome? If so, is this surprising? What might be the mechanism for an independent paternal genotype effect on PW? Also clarify if this analysis (in lines 239 to 243) is restricted to the trios?

RESPONSE

The genomic SEM analysis, from which this estimate derives, produces estimates for fetal, maternal, and paternal SNP heritability conditional on the other genomes, so this estimate is conditional on both fetal and maternal genome. We agree that non-zero estimate for the paternal genome is initially surprising, but it may be driven by non-additive genetic effects not accounted for in the model (e.g. parent-of-origin effects). We have added this to the manuscript (paragraph 1, page 10).

The method is not restricted to trios and uses the full meta-analysis summary statistics. We have added text to the paper to clarify this (paragraph 1, page 25).

Lines 318 to 328 Correlations between PW and BW - this section is unnecessarily long and confuses BW vs PW comparisons with fetal vs parental comparisons which have already been described. Restrict to PW vs BW correlations for each genome source.

RESPONSE

Thank you for drawing our attention to this, we have simplified the text (paragraph 1, page 13). However, we believe it is valuable to compare fetal effects on PW with maternal effects on BW (and *vice versa*). These are relevant comparisons to make in the context of placenta sacrifice over the fetal needs (Broad et al., 2011, PMID: 21810990).

Figures 3a and 5 are confusing and do not support the results, likely because they both unhelpfully combine fetal vs parental AND 'PW-only' vs 'PW & BW' comparisons:

RESPONSE

We agree that the clarity of Figures 3a and 5 could be improved and have now refactored them completely. As suggested by the reviewer, we have tried to better distinguish the fetal vs. parental and PW vs. BW comparisons. We think that the new versions are much clearer and thank the reviewer for this suggestion.

Fig 3a - Is meant to demonstrate the discrimination between fetal vs parental effects but does not achieve this. For some 'fetal' genes (e.g. RPL31P11, ARHGAP26, LOC339593) the MT (fetal) effect does not seem larger than the MnT (maternal) effect. CHRM3 (labelled 'ambiguous') appears to have a much larger MT than MnT effect. Furthermore, the classification of loci as PW-only or 'PW & BW' appears to be inconsistent with the displayed effects on PW and BW - possibly because this MoBa-only sample was not used for this classification (Fig 3 is not cited in that description on page 13).

RESPONSE

As suggested by the reviewer, we have now refactored Figure 3A to focus on the fetal vs. parental comparison. The lines are now clustered accordingly and we have removed the BW analyses in MoBa. We chose to keep the BW allele transmission analysis by Juliusdottir et al., 2021 (PMID: 34282336) and maternal and fetal analyses by Warrington et al., 2019 (PMID: 31043758) because they provide valuable information on the consistency of fetal vs. parental relative effects between the two phenotypes. The origin of the values and reason for missingness are better highlighted on the figure and detailed in the legend. To facilitate the understanding of the figure, the columns have been reordered according to their order of appearance in the text: PW meta-analysis, PW allele transmission analysis, and symmetrically for BW. We have also further emphasized the evidence (*P*-value) of the association by scaling the size of the square according to the *P*-value at the association test. In this way, we believe that the results with higher certainty will be more clearly visualized compared to minor non-significant statistical fluctuations.

The reviewer is correct that the fetal vs. maternal classification was done on the entire cohort and is therefore not fully mirrored in the MoBa data. We have inspected the ‘fetal’ SNP, and especially the loci pointed by the reviewer: *RPL31P11*, *ARHGAP26*, *LOC339593*. In the absence of a paternal effect and of a parent-of-origin effect, the fetal effect will be carried by MT and PT, and the maternal effect by MnT and MT. In Figure 3a, for the SNPs classified as ‘fetal’, the two transmitted alleles show a positive association, while the maternal non-transmitted allele generally presents a more modest effect in the opposite direction (see Figure B below where these loci are illustrated). This is consistent with a fetal effect, and when MnT carries an opposite effect, the fact that MT is consistent with PT indicates that the fetal effect dominates. This is the case for *RPL31P11* and *ARHGAP26*, where both PT and MT have positive effects and the fact that MT is not influenced by the seemingly negative effect on MnT is consistent with a fetal effect. This can be better observed in Supplementary Figure 5. For *RPL31P11* and *ARHGAP26*, MnT presents a negative point estimate, while both transmitted alleles show consistent positive effects; this results in a confident child effect but less conclusive maternal effect in the opposite direction, which did not reach our selection criterion in the meta-analysis. The absence of BW-transmission results for these two markers in Juliusdottir et al., 2021 (PMID: 34282336) indicates that there are no GWA-significant BW-associated markers in high LD with the PW-SNPs. For *LOC339593*, all alleles show positive effect in MoBa, with stronger effects for the transmitted alleles, which would indicate a fetal effect possibly combined with a maternal effect in the same direction - here again the maternal effect was not conclusive in the meta-analysis and the locus is therefore classified as fetal - a result that is in line with the BW transmission pattern for the same marker in Juliusdottir et al., 2021 (PMID: 34282336). For *CHRM3*, the effect in MoBa indeed seems to support a predominant fetal effect, but the large confidence intervals did not allow us to ascribe this one to a particular category.

Figure B: Zoom-in of refactored Figure 3A (of the main paper)

As the reviewer pointed out, we aimed at depicting the BW vs. PW comparison in Figure 5. The BW values presented in Figure 3A rather aim at showing how the transmission-resolved effects align between PW and BW, as illustrated with the striking agreement in direction of effects for MnT, MT, and PT between PW and BW in MoBa and the analysis by Juliusdottir et al., 2021 (PMID: 34282336), respectively. We hope that the new layout of the figures better convey this point.

One thing that the reviewer will notice however, is that most variants where transmission-resolved effect sizes are provided by Juliusdottir et al., 2021 (PMID: 34282336) fall in PW & BW categories, while the vast majority of SNPs classified as 'Predominantly PW' do not have proxies in the BW analysis of Juliusdottir et al, 2021 (PMID: 34282336) that only presented transmission results for markers that reach GWA significance in their meta-analysis. We argue that this is consistent with these variants being predominantly specific to PW.

Fig 5 is also too complex and fails to demonstrate the discrimination between the 12 PW-only signals and the 28 'PW & BW' signals. Addition of fetal vs parental comparisons confuses this aim.

RESPONSE

We have attempted to simplify the figure by only plotting fetal effects, and focussing on the BW/PW classification.

Fig 4 - Clarify that the Y axis label indicates "Rg with PW"

RESPONSE

Thank you for the suggestion. It has been implemented.

Fig 6 - It seems that the Y-axis labels for maternal and fetal genomes are erroneously swapped.

RESPONSE

Thank you for pointing this out. It has been corrected.

Line 380 - the 31 fetal PW genes ranked higher (than other genes) in maternal innate lymphocyte cells ($P = 1.2 \times 10^{-3}$). What is the explanation for this - i.e. how can fetal genotype alter gene expression in maternal cells?

RESPONSE

The results do not imply that fetal genotype alters gene expression in maternal cells, merely that genetic variation which, when present in the fetal genome affects PW, lies close to genes, which are highly expressed in maternal innate lymphocyte cells.

Interesting functional insights are provided from analyses of mRNA expression and methylation. It would be helpful to cite a reference for the stated role of SLC7A5 in the placenta (Line 415)

RESPONSE

We have added a relevant reference to the text as indicated (ref. 35).

Line 508 - PW and later neuropsychiatric traits - cite evidence for the link between these traits, and state which direction has been reported? Is the observed relationship with ADHD consistent with literature? Are genetic correlations seen? Line 512-515, confirmation of the PW-PRS association with PW in these patient cohorts is expected and could be moved to the Methods.

RESPONSE

Our motivation for these analyses is based on a general hypothesis about fetal origins of mental health (Barker, 2007, PMID: 17444880). Many epidemiological studies have reported associations between lower BW and a higher risk of psychiatric disorders later in life, including e.g. schizophrenia (Abel et al., 2010, PMID: 20819986), depression (Colman et al., 2012, PMID: 22762297), and attention deficit/hyperactivity disorder (ADHD) (Rahman et al., 2021, PMID: 34556092). However, it is not clear what the causal factors are that underlie these associations.

Placental dysfunction has been associated with adverse neurodevelopmental outcomes (e.g. Kratimenos et al., 2019, PMID: 31003234, Leon et al., 2022, PMID: 33864014), which is why we used the opportunity to analyze data from the iPSYCH study for potential associations of observed PW as well as PGS for PW with later neuropsychiatric traits. While many epidemiological studies have analyzed BW and later mental health, PW has received little attention. One study (Khalife et al., 2012, PMID: 22792364) reported associations between increased PW (adjusted for GA, BW, socio-demographic factors, and medical factors) and higher risk of teacher or parent-reported ADHD symptoms in boys, but did not assess mental health diagnoses. We found a small nominally significant association between higher observed PW and reduced risk of an ADHD diagnosis. The associations between observed BW, PW, and later mental health warrant further epidemiological study, e.g. based on large national registers available in Nordic countries, but that is beyond the scope of the current paper. We have added references to articles about placental dysfunction and adverse neurodevelopmental outcomes (Kratimenos et al., 2019, PMID: 31003234; Leon et al., 2022, PMID: 33864014) and have also included genetic correlation results between PW and the analyzed neuropsychiatric traits in supplementary Table 19 (all were very small and non-significant). We include the reference to the PW-PRS supplementary figure in the Results, as it shows that patterns are as expected, also in the groups of cases for the diseases.

Line 542-559 - Discussion on KCNQ1 ignores the finding here that this signal is classified as 'PW-only'. Instead this section rationalises for various effects of KCNQ1 on both fetal and placental growth. This should be corrected / explained.

RESPONSE

Thank you for bringing this overlook to our attention, the discussion has been extended accordingly (paragraph 1, page 23).

Line 573-577 - FLT1 is not related to PW so has low relevance here. This section should try to rationalise why higher PW is associated with preeclampsia risk.

RESPONSE

Thank you for the suggestion. We agree and have now removed the text about *FLT1* and have edited paragraph 2, pages 23 to explain how our PW-preeclampsia result is consistent with the hypothesis that late-onset preeclampsia may result from intervillous overcrowding as placental growth reaches its limits at term.

Line 747 - Enrichment analysis Methods "Significance levels for this enrichment analysis were Bonferroni corrected...". State the corrected thresholds.

RESPONSE

The Bonferroni thresholds have been added to the text (paragraph 1, page 29).

Decision Letter, first revision:

14th Jun 2023

Dear Pål,

Thank you for submitting your revised manuscript "Genome-wide association study of placental weight in 65,405 newborns and 113,620 parents reveals distinct and shared genetic influences between placental and fetal growth" (NG-A61055R1). It has now been seen by the original referees and their comments are below. The reviewers find that the paper has improved in revision, and therefore we'll be happy in principle to publish it in Nature Genetics, pending minor revisions to satisfy the referees' final requests and to comply with our editorial and formatting guidelines.

Sincerely,

Michael Fletcher, PhD
Senior Editor, Nature Genetics
ORCID: 0000-0003-1589-7087

Reviewer #1 (Remarks to the Author):

I have been through the responses in some detail. Whilst I do not agree entirely with the authors' conceptual approach, I feel that they have adequately responded to the criticisms. In the circumstances, therefore, I would recommend publication.

Reviewer #3 (Remarks to the Author):

The authors have responded appropriately to my comments.

I have some further comments following their responses to Reviewer 1:

They state there may be "potentially interesting association signals with PW that are mediated by parity". A simple test of this would be to explore the identified PW signals for association with parity.

Page 12 - "Correlations between PW and BW: An interesting observation is in Figure 4c where the fetal effects from the WLM on BW have a negative genetic correlation with the maternal effects from the WLM on PW." The authors note in their reply to Reviewer 1 the dangers of conditional analyses. Could this negative correlation just reflect overadjustment, i.e. Collider bias? Please add this as a possibility.

Author Rebuttal, first revision:

RESPONSES

Editor:

See comments and edits to the Author Checklist, Reporting Summary, manuscript, figures, tables, and supplementary material.

Reviewer #1:

Comment:

I have been through the responses in some detail. Whilst I do not agree entirely with the authors' conceptual approach, I feel that they have adequately responded to the criticisms. In the circumstances, therefore, I would recommend publication.

Response:

Thanks!

Reviewer #3 (Remarks to the Autho:

Comment 1:

They state there may be "potentially interesting association signals with PW that are mediated by parity". A simple test of this would be to explore the identified PW signals for association with parity.

Response 1:

We thank the reviewer for this suggestion. However, we think that the comment may have been taken out of context. We assume that the Reviewer is referring to our previous response to Reviewer 1, comment 1. In that response, we did not state that "there may be potentially interesting association signals with PW that are mediated by parity". Rather, we stated, "As the reviewer points out, conditioning on another phenotype like parity would reduce phenotypic variation, hence increasing power to detect genetic association, but it would also remove potentially interesting association signals with PW that are mediated by parity. In general, adjusting for heritable traits is known to be a source of bias in GWA studies, and it is recommended to keep such covariates to a minimum (Aschard et al., 2015, PMID: 25640676)."

So our comment about potentially interesting associations was made in the context of explaining why we chose not to adjust for parity (or indeed several other potential mediators). Furthermore, in that response to Reviewer 1, we went on to show that none of the association signals were changed on adjustment for parity (consistent with them not being mediated by parity). We include that data again here for reference (Fig. A).

Figure A: Effect size estimates and 95% confidence intervals for the association with PW of the different signals in 19,861 unrelated trios in MoBa using different sets of covariates in addition to the batches and principal components: (1) none, (2) sex, (3) sex + GA, (4) sex + GA + parity

Covariates → none sex sex + ga sex + ga + parity

Comment 2:

Page 12 - "*Correlations between PW and BW: An interesting observation is in Figure 4c where the fetal effects from the WLM on BW have a negative genetic correlation with the maternal effects from the WLM on PW.*" The authors note in their reply to Reviewer 1 the dangers of conditional analyses. Could this negative correlation just reflect overadjustment, i.e. Collider bias? Please add this as a possibility.

Response 2:

We have added the point as a possibility in the Results (Section *Correlations between placental weight and birth weight*, last sentence).

Final Decision Letter:

31st Aug 2023

Dear Pal,

I am delighted to say that your manuscript "Genome-wide association study of placental weight identifies distinct and shared genetic influences between placental and fetal growth" has been accepted for publication in an upcoming issue of Nature Genetics.

Your paper will be published online after we receive your corrections and will appear in print in the next available issue. You can find out your date of online publication by contacting the Nature Press Office (press@nature.com) after sending your e-proof corrections. Now is the time to inform your Public Relations or Press Office about your paper, as they might be interested in promoting its publication. This will allow them time to prepare an accurate and satisfactory press release. Include your manuscript tracking number (NG-A61055R2) and the name of the journal, which they will need when they contact our Press Office.

Please note that *Nature Genetics* is a Transformative Journal (TJ). Authors may publish their research with us through the traditional subscription access route or make their paper immediately open access through payment of an article-processing charge (APC). Authors will not be required to make a final decision about access to their article until it has been accepted. > Find out more about Transformative Journals

Authors may need to take specific actions to achieve compliance with funder and institutional open access mandates. If your research is supported by a funder that requires immediate open access (e.g. according to Plan S principles) then you should select the gold OA route, and we will direct you to the compliant route where possible. For authors selecting the subscription publication route, the journal's standard licensing terms will need to be accepted, including https://www.nature.com/nature-portfolio/editorial-policies/self-archiving-and-license-to-publish. Those licensing terms will supersede any other terms that the author or any third party may assert apply to any version of the manuscript.

An online order form for reprints of your paper is available at https://www.nature.com/reprints/author-reprints.html. Please let your coauthors and your institutions' public affairs office know that they are also welcome to order reprints by this method.

If you have not already done so, we invite you to upload the step-by-step protocols used in this manuscript to the Protocols Exchange, part of our on-line web resource, natureprotocols.com. If you complete the upload by the time you receive your manuscript proofs, we can insert links in your article that lead directly to the protocol details. Your protocol will be made freely available upon publication of your paper. By participating in natureprotocols.com, you are enabling researchers to more readily reproduce or adapt the methodology you use. [Natureprotocols.com](http://natureprotocols.com) is fully searchable, providing your protocols and paper with increased utility and visibility. Please submit your protocol to <https://protocolexchange.researchsquare.com/>. After entering your nature.com username and password you will need to enter your manuscript number (NG-A61055R2). Further information can be found at <https://www.nature.com/nature-portfolio/editorial-policies/reporting-standards#protocols>

Sincerely,

Michael Fletcher, PhD
Senior Editor, Nature Genetics

ORCID: 0000-0003-1589-7087